🔓 | **Open Peer Review** | Applied and Industrial Microbiology | Research Article

# Oxytetracycline hyper-production through targeted genome reduction of *Streptomyces rimosus*

Alen Pšeničnik,[1] Lucija Slemc,[1] Martina Avbelj,[1] Miha Tome,[1] Martin Šala,[2] Paul Herron,[3] Maksym Shmatkov,[4,5] Marko Petek,[6] Špela Baebler,[6] Peter Mrak,[7] Daslav Hranueli,[4] Antonio Starčević,[4] Iain S. Hunter,[3] Hrvoje Petković[1]

**ABSTRACT** Most biosynthetic gene clusters (BGC) encoding the synthesis of important microbial secondary metabolites, such as antibiotics, are either silent or poorly expressed; therefore, to ensure a strong pipeline of novel antibiotics, there is a need to develop rapid and efficient strain development approaches. This study uses comparative genome analysis to instruct rational strain improvement, using *Streptomyces rimosus*, the producer of the important antibiotic oxytetracycline (OTC) as a model system. Sequencing of the genomes of two industrial strains M4018 and R6-500, developed independently from a common ancestor, identified large DNA rearrangements located at the chromosome end. We evaluated the effect of these genome deletions on the parental *S. rimosus* Type Strain (ATCC 10970) genome where introduction of a 145 kb deletion close to the OTC BGC in the Type Strain resulted in massive OTC overproduction, achieving titers that were equivalent to M4018 and R6-500. Transcriptome data supported the hypothesis that the reason for such an increase in OTC biosynthesis was due to enhanced transcription of the OTC BGC and not due to enhanced substrate supply. We also observed changes in the expression of other cryptic BGCs; some metabolites, undetectable in ATCC 10970, were now produced at high titers. This study demonstrated for the first time that the main force behind BGC overexpression is genome rearrangement. This new approach demonstrates great potential to activate cryptic gene clusters of yet unexplored natural products of medical and industrial value.

**IMPORTANCE** There is a critical need to develop novel antibiotics to combat antimicrobial resistance. *Streptomyces* species are very rich source of antibiotics, typically encoding 20–60 biosynthetic gene clusters (BGCs). However, under laboratory conditions, most are either silent or poorly expressed so that their products are only detectable at nanogram quantities, which hampers drug development efforts. To address this subject, we used comparative genome analysis of industrial *Streptomyces rimosus* strains producing high titers of a broad spectrum antibiotic oxytetracycline (OTC), developed during decades of industrial strain improvement. Interestingly, large-scale chromosomal deletions were observed. Based on this information, we carried out targeted genome deletions in the native strain *S. rimosus* ATCC 10970, and we show that a targeted deletion in the vicinity of the OTC BGC significantly induced expression of the OTC BGC, as well as some other silent BGCs, thus suggesting that this approach may be a useful way to identify new natural products.

**KEYWORDS** genome reduction, antibiotic biosynthesis, oxytetracycline, cryptic metabolites

Address correspondence to Hrvoje Petković, hrvoje.petkovic@bf.uni-lj.si.

University of Ljubljana Biotechnical faculty filed a patent application relating to the findings of this work (inventors: Hrvoje Petković, Alen Pšeničnik, and Lucija Slemc).

See the funding table on p. 27.

Antimicrobial resistance (AMR) presents one of the greatest challenges for modern health care, and there is a critical need to develop novel antibiotics to combat AMR. Microorganisms are still a rich source of antibiotics (1), and it is estimated that

only 3% of the natural products encoded in bacterial genomes have been characterized experimentally (2). Under laboratory conditions, most of the biosynthetic gene clusters (BGC) encoding biosynthesis of microbial secondary metabolites are either silent or expressed poorly so that their products are only detectable at nanogram quantities in crude extracts. Novel compounds with good preclinical potential often cease development because they cannot be produced in sufficient quantities at reasonable cost and time invested relative to the commercial return (3, 4). Successfully mining the wealth of potentially bioactive molecules requires advanced technologies to produce ample quantities of natural products (and their derivatives, analogs, and mimetics) in multigram quantities. Strain improvement is the cornerstone of both drug discovery and any subsequent commercial fermentation process.

Over the last 80 years, new and more efficient industrial microbial strains and processes have been launched, resulting in purer and more affordable products (5). Throughout this period, strain improvement has resulted almost exclusively from essentially random, empirical, labor-intensive, and time-consuming approaches of mutagenesis and strain selection (6). Modern knowledge-based scientific methods such as metabolic engineering and synthetic biology are now available for rational strain improvement and have been used successfully with microorganisms such as *Escherichia coli* and *Corynebacterium glutamicum* (7–9) for overproduction of amino acids and vitamins with simple production pathways. By contrast, rational approaches to improve titers of antibiotics have been less successful—perhaps due to the complexities of secondary metabolism and the associated antibiotic molecular structures. Various rational approaches to titer improvement of secondary metabolites have been developed such as (over)expression of target BGC in the native or heterologous hosts (10, 11), modulation of general or pathway-specific regulatory genes (12, 13), upregulation of biosynthetic pathways providing building blocks (substrates) for biosynthesis of target metabolites (14), debottlenecking of biosynthetic pathway steps (15), increasing antibiotic export and resistance mechanisms (12), and removal of competing pathways involved in biosynthesis of undesired metabolites (impurities), which can improve both quality and the final titer of the target product (16, 17). The recent development of novel *Corynebacterium glutamicum* strains for hyperproduction of amino acids benefited greatly from a systems metabolic engineering approach (9) that used comparative analyses of the genomes (and other omics-data) to determine how industrial high-titer strains, which were generated over decades by repeated random mutagenesis and selection, evolved from their progenitors (parental wild-type strains). Guided by this, we investigated the evolution of an antibiotic production lineage. This study uses comparative genome analysis to instruct rational strain improvement, using *Streptomyces rimosus* as a model strain. Over the last 70 years, *S. rimosus* has been used for the industrial production of the medically important broad-spectrum antibiotic oxytetracycline and high-titer strains have been developed achieving over 35 g/L (18). The biosynthesis of oxytetracycline has been studied in detail (19), while the genome of *S. rimosus* ATCC 10970 (the original soil isolate) has been sequenced and consists of one linear chromosome of around 9 Mb and one linear plasmid of around 300 kb (20–22). We had access to strains from two pharmaceutical companies that had conducted independent empirical strain improvement programs to commercialize oxytetracycline. These were strains M4018 (Pfizer) (23) and R6 (PLIVA) (24, 25), both derived independently from *S. rimosus* ATCC 10970. We have now completed high-quality sequencing of the entire genomes of M4018 and R6-500 and compared their genomes to the native isolate *S. rimosus* ATCC 10970. During analysis of the genomes of M4018 and R6, we identified large deletions and complex DNA rearrangements located at the terminal regions of the linear chromosomes in the vicinity of the OTC BGC, which is located around 600 kb from one chromosomal end. As the DNA rearrangements occurred in approximately the same location in these two independently derived strains, we evaluated the potential effect of these large rearrangements on OTC production. Using an optimized CRISPR-Cas9 tool (26), precise 145 kb and 240 kb deletions were introduced in the vicinity of the

OTC BGC towards the end of the chromosome of the ATCC 10970 Type Strain and the transcriptomic and metabolomic effects of these deletions were determined, as well as their effect on OTC productivity. Surprisingly, a single genome engineering step, in which we introduced a 145 kb deletion, resulted in a remarkable increase of the OTC titer. To investigate further the potential effects of these deletions on OTC biosynthesis, we carried out comparative whole-genome transcription analysis, which clearly demonstrated massive overexpression of the genes present in the OTC BGC. This implies that enhanced transcription of the OTC BGC is responsible for the increase in OTC titer and not enhanced substrate supply. Furthermore, by chromosomal integration of a second copy of the OTC gene cluster, an additional significant increase in OTC titer was achieved, suggesting that expression of the OTC cluster is the main driver of titer increase in industrial strains. Surprisingly, we also observed changes in expression of other BGCs, independently of their chromosomal location. *S. rimosus* ATCC 10970 contains over 45 putative BGCs. Analysis of metabolites produced by the engineered strain containing the 145 kb deletion showed changes in secondary metabolites profile. Some metabolites, previously undetectable by LC/MS in *S. rimosus* ATCC 10970, were now produced at relatively high titer. This approach demonstrates that a single deletion greatly increases the titer of the main antibiotic produced and activates so-called cryptic gene clusters, which presents an enormous source of yet unexplored natural products of medical and industrial value.

## RESULTS

### Comparative bioinformatic analysis of genomes of industrial *Streptomyces rimosus* strains M4018 and R6 with the parent strain ATCC 10970

As both M4018 and R6-500 strains had significantly greater oxytetracycline titers than the ancestral strain ATCC 10970, we first sought to determine the genomic changes that had occurred during the strain improvement programs carried out by Pfizer and PLIVA, respectively. We generated complete genome sequences of M4018 and R6-500 (see Supplementary Information 1 for details of these assemblies and deduced rearrangements) and noticed that both strains had undergone extensive rearrangements in both left-hand end (LHE) and right-hand end (RHE) (Fig. 1A and B). Detailed comparison of the chromosomes is displayed in Fig. S1. BGCs present in M4018 and R6-500 were identified by AntiMSASH 6.0 (27) and compared (Table S1) with BGCs identified previously in ATCC 10970 (21). In *Streptomyces* species, the chromosomal arms are abundant with biosynthetic gene clusters (28). The genomic alterations observed had resulted in extensive "shuffling" of the repertoire of BGCs and their locations in these industrial strains. For example, M4018 had lost 78,065 bp from the LHE of the ATCC 10970 chromosome (genes SRIM_000005 to SRIM_000425) and lost 284,432 bp (genes SRIM_038260 to SRIM_039170) from the RHE, adjacent to the OTC BGC, which was now located almost at the end of right chromosomal arm (Fig. 1D; Fig. S1). The terminal 323,390 bp from the RHE of the ATCC 10970 chromosome (genes: SRIM_039175 to SRIM_040530) had fused to the LHE of the M4018 chromosome. In this way, BGCs 5–8 were deleted from the RHE and BGCs 1–4 were relocated to the opposite chromosomal arm in an inverted orientation (marked yellow on Fig. 1D).

In R6-500, the terminal 869,667 bp from the RHE of the ATCC 10970 chromosome (genes: SRIM_037035 to SRIM_040530) had duplicated and fused to the LHE of the chromosome (Fig. S1). 555,093 bp (genes: SRIM_000005 to SRIM_002375) were lost from the LHE of the ATCC 10970 chromosome during the origination of R6-500. Subsequently, a deletion had occurred between gene positions SRIM_036310 and SRIM_038540. Consequently, eight BGCs were deleted in R6-500: BGCs 11–13 and 40–46 (Table S1, Supplementary Information 2). Furthermore, the R6-500 genome now contained two copies of the DNA fragment encompassing BGCs 1–7 (Fig. 1E; Fig. S1). Although not duplicated, the OTC cluster is found at the LHE of the R6-500 chromosome as opposed to the RHE of ATCC 10970. The linear plasmid has undergone some minor deletions in M4018, whereas ~30% of the LHE has been deleted in R6-500 (21).

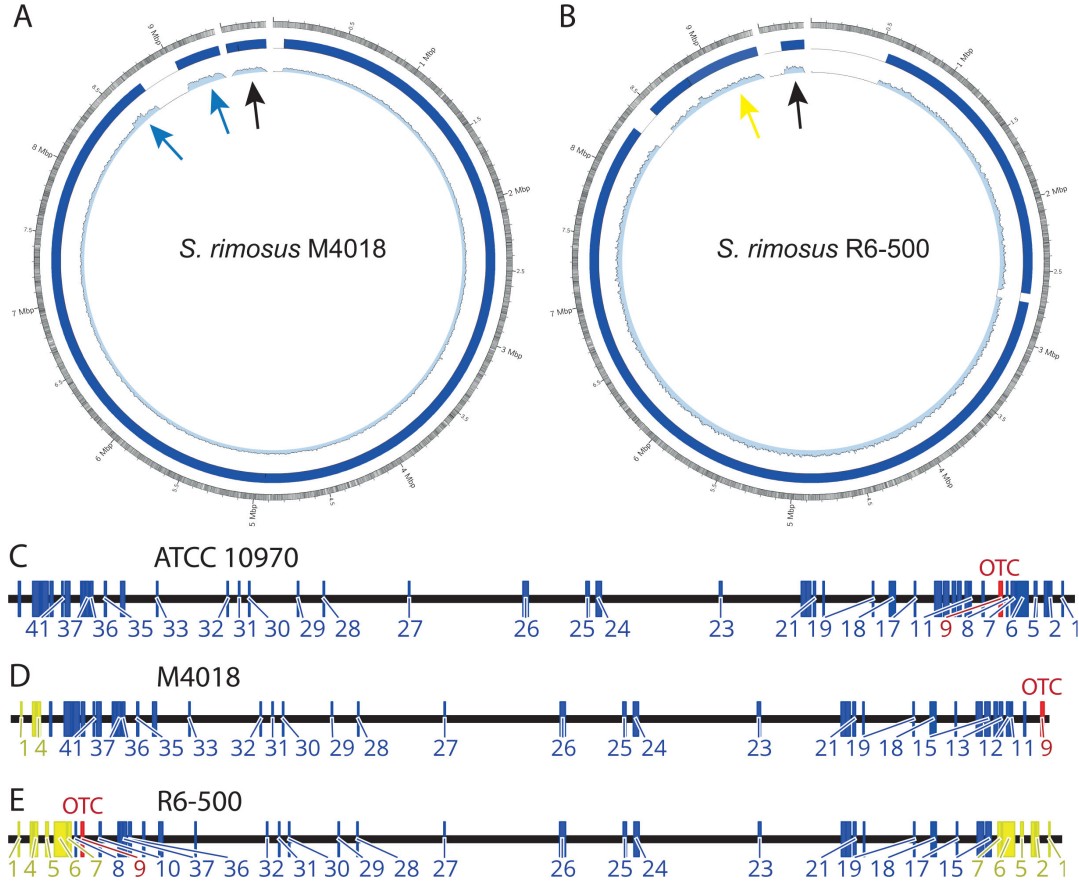

**FIG 1** Circos plots of Icarus contig alignments of *S. rimosus* M4018 (A) and R6-500 (B) assemblies against *S. rimosus* ATCC 10970 reference and comparison of the location of BGCs on the chromosome of *S. rimosus* ATCC 10970 (C), M4018 (D), and R6-500 (E) strains. Chromosome and plasmid (black arrows) assemblies were evaluated with ATCC 10970 set as the reference sequence. Outer circle—ATCC 10970 chromosome and plasmid, inner circle—sequence coverage of R6-500 (A) and M4018 (B). Terminal regions of M4018 that we suspect to be duplicated due to double read-depth of PacBio sequences are displayed (blue arrows), while the terminal inverted repeat of R6-500 is also indicated (yellow arrow); this is present at both ends of our R6-500 assembly. The location of the OTC BGC (No. 9) is marked in red. The duplicated region at the chromosome arms of strain R6-500 (E) and translocated region in M4018 (D) are marked in yellow.

## Identification of the chromosomal regions for targeted deletion experiments in *S. rimosus* ATCC 10970

Our genome analysis showed that both M4018 and R6-500 had large-scale rearrangements at one end of their linear chromosomes, centered around the OTC BGC. We had already observed that relocation of the OTC BGC to a different chromosomal location had a positive effect on OTC production (29). We therefore tested whether changing the location of the OTC BGC relative to the end of the chromosome had a positive effect on OTC production. This was done by introducing a large deletion in ATCC 10970, using the deletion observed in M4018 as a guide. We evaluated two large deletions with sizes of 145 and 240 kb (Fig. 2). The 145 kb deletion removed two BGCs (6 and 7) from the ATCC 10970 genome (strain ATCC Δ145kb). The deletion of the 240 kb region removed BGC 5 in addition to BGCs 6 and 7 (strain ATCC Δ240kb).

BGC 6 (Fig. 2, which is alongside the OTC BGC) encodes biosynthesis of rimocidin (a major metabolite in *S. rimosus*), a polyene macrolide with antifungal activity (30). In our recent study, we demonstrated that ATCC 10970 produces not only rimocidin but also rimocidin congeners CE-108 and amid-CE-108 (21), which are biosynthesized by a PKS type I enzyme complex (31) containing 14 modules, making it one of the largest BGCs present in the genome of *S. rimosus*. Based on AntiSMASH analysis (27), BGC 7 encodes an NRPS gene cluster for a yet unknown secondary metabolite (Fig. S2;

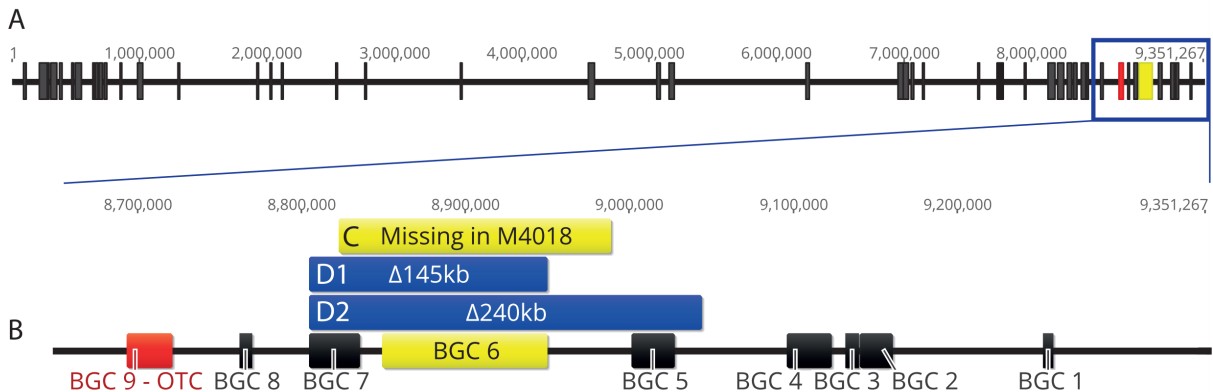

**FIG 2** Schematic presentation of the end of the chromosome containing OTC BGC in *S. rimosus* ATCC 10970. (A) Chromosome of ATCC 10970 with marked BGCs (Table S1). (B) Right end of the chromosome, containing OTC BGC. The extent of the deletion present in M4018 is shown (yellow) (C), and the locations of the deletions introduced in engineered strains of ATCC Δ145kb and ATCC Δ240kb are in blue (D1 and D2).

genes presented in Table S2). BGC 5, removed additionally in ATCC Δ240kb, encodes biosynthesis of a yet unknown PKS type I metabolite. Table S2 lists genes deleted in ATCC Δ145kb (SRIM_038560-SRIM_038870), including genes of BGC 6 (rimocidin) and BGC 7 (unknown metabolite) and the additional genes deleted in ATCC Δ240kb (SRIM_038560-SRIM_039255), which includes those of BGC 5. The AntiSMASH analysis of BGCs 5, 7, and 22_2 is presented in Fig. S2.

## Deletion of chromosomal regions adjacent to the OTC BGC in *S. rimosus* ATCC 10970

Using pREP_P1_cas9_tsr as a foundation (26, 32), two plasmid constructs (pRep_P1_cas9_Δ145kb and pRep_P1_cas9_Δ240kb) were used to introduce deletions into the ATCC 10970 chromosome (Supplementary information 3). They contained two successive gRNA cassettes, designed to target both ends of the deleted regions, along with adjacent homologous regions of approximately 2 Kbp in size (Table S3). Plasmids were introduced to *S. rimosus* ATCC 10970 via conjugation (see Materials and Methods). The overall procedure for creating the deletions in *S. rimosus* using CRISPR-Cas9 had been optimized previously using the Cas9-SD-GusA tool (26). The deletion of the 145 kb region was carried out in ATCC 10970 and ATCC 10970 ΔOTC, already containing a deletion of the OTC cluster (29). Verification of PCRs on colony isolates confirmed that the anticipated deletion events had occurred (Fig. S3; Supplementary information 4; Table S4). We next performed detailed analysis of the best-performing ATCC strain with the 145 kb deletion, designated ATCC Δ145kb, but another independent clone with an introduced 145 kb deletion (designated ATCC Δ145kb-b) was also included in OTC fermentations and transcriptome analysis (Fig. S4).

## Comparative analysis of OTC production in *S. rimosus* M4018, R6, ATCC 10970 and engineered strains with 145 and 240 kb deletions

OTC production of the best-performing *S. rimosus* mutants with 145 and 240 kb deletions from a pre-screening experiment was compared with ATCC 10970 and the industrial OTC-producing strains M4018 and R6-500. Remarkably, ATCC Δ145 kb had a significant increase in OTC production of 2.9 g/L—a remarkable 7-fold increase compared to 0.452 g/L produced by its ATCC 10970 parent. This is comparable with the range of titers of the industrial production strains M4018 (2.3 g/L) and R6-500 (1.2 g/L) (Fig. 3) derived by intensive strain selection. ATCC Δ240kb produced 0.844 g/L OTC, still significantly higher than its parental strain ATCC 10970 (Fig. 3). These increases in OTC titers were notable as the growth rates (based on DNA concentration measurements throughout the fermentation process) of the engineered strains were comparable to parental ATCC

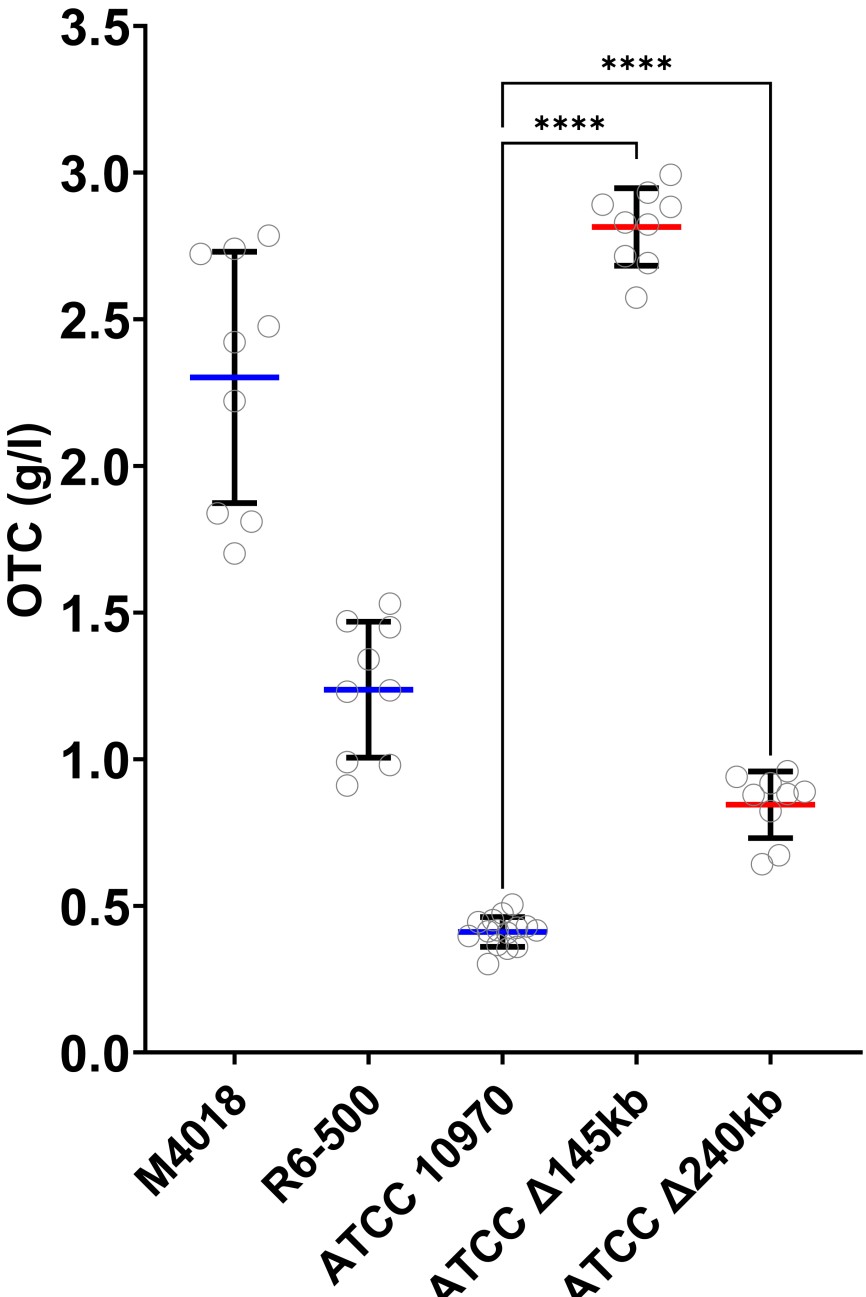

**FIG 3** Production of OTC by industrial strains M4018 and R6-500 and engineered strains with 145 and 240 kb deletion compared to the parent strain *S. rimosus* ATCC 10970. Mean with error bars showing s.d. (*n* = 9, three independent fermentations from three biological replicates). Significance is tested with Dunnett's T3-test, \*\*\*\**P* < 0.0001.

10970 strain (Fig. S5). Results of OTC production for another independently isolated strain with 145 kb deletion (ATCC Δ145kb-b) are presented in Fig. S4A.

Alongside significant differences in OTC titers, ATCC Δ145 kb but not ATCC Δ240 kb also displayed an unusual morphology compared to the ATCC 10970 strain when cultivated on SM agar plates (Fig. S6). Most notably, ATCC Δ145 kb had aerial mycelium of the "peeling" phenotype, not observed with ATCC 10970 (Fig. S6A and B). On the other hand, we did not detect large differences in growth rates during the evaluation of growth characteristics in OTC seed medium (GOTC-V) and production medium (GOTC-P) (Fig. S5). However, culturing of ATCC Δ145 kb decreased the pH more rapidly in both

vegetative and production medium compared to other strains, indicating changes in primary metabolism in comparison to the Type Strain (Fig. S5B).

## Confirmation of targeted deletion of the selected DNA fragment by entire-genome sequencing

Mapping of Illumina reads to the S. *rimosus* ATCC 10970 reference genome (21) confirmed that the anticipated 145 kb region had been deleted in the ATCC Δ145 kb strain (also in the second strain-ATCC Δ145 kb-b; Supplementary Information 5). There was a complete absence of sequencing reads internal to the region of ATCC 10970 that had been deleted. To ensure reliable identification of possible non-specific mutations, two control samples of the ATCC 10970 parental strain were also re-sequenced. The in-depth WGS analysis of the ATCC Δ145kb strain revealed only a few non-specific mutations per genome (Table S5), suggesting that despite invasive perturbation of the genome to create the 145 kb deletion, there was little CRISPR-Cas9 "off target" activity in the engineered strain.

## Targeted inactivation of BGC encoding rimocidin biosynthesis

The entire rimocidin BGC (BGC 6, see Fig. 2) was removed in both ATCC Δ145 kb and ATCC Δ240 kb strains. It was plausible that the increased OTC titers observed with these deletion strains might be attributed to increased availability of precursors (both OTC and rimocidin require malonyl-CoA[14]) or some cross-regulation issue between the OTC and rimocidin BGCs. Seco et al. (31) proposed that inactivation of rimocidin biosynthesis in *Streptomyces diastaticus* var.108 had a significant effect on OTC biosynthesis. Therefore, to clarify the potential effect of deletion of the rimocidin BGC on OTC production by the Δ145kb strain, we introduced a small in-frame deletion inside the rimocidin BGC in ATCC 10970 that specifically inactivated the loading module, RimA. Similarly, although in a different *Streptomyces* species, Seco et al. (31) also inactivated *rim*A via insertional inactivation. We constructed a CRISPR plasmid pRep_P1_cas9_ΔrimA (see Materials and Methods), specifically targeting the ACP (acyl carrier protein) and KS (encoding ketosynthase) domains of the loading module RimA (Fig. S3C). The deletion of the target DNA fragment was confirmed by PCR (Fig. S3C), and the absence of all rimocidin-related metabolites (RIM) was confirmed by full-scan LC-MS (see Fig. 9). Inactivation of *rimA* and, therefore, interruption of rimocidin biosynthesis had no effect on OTC titer. This was confirmed by HPLC analysis of fermentation broths from four independent *rimA* mutants (mean ATCC 10970 = 0.483 g/L, mean all *rimA* mutants = 0.402 g/L). For direct comparison, we included ATCC Δ145 kb in this fermentation experiment, which on average produced 3.2 g/L (Fig. 4).

## Comparative transcriptional analysis of *S. rimosus* ATCC 10970 and engineered strains containing deletions of the 145 and 240 kb regions

To elucidate the observed changes in OTC production and morphology with genetic modifications, we have performed differential gene expression analysis of the wild type and the two engineered strains at two time points, 24 and 50 h, corresponding to the onset of production and late exponential stage (Fig. S7). Cultivation procedure and isolation of mRNA are described in Supplementary Information 6. In line with OTC titers (Fig. 3) and strain morphology (Fig. S6), we observed pronounced differences in the transcriptome of ATCC Δ145kb compared to ATCC Δ240kb (Fig. 5 and 6; Fig. S8). This is somewhat surprising considering that the Δ240kb strain had lost a larger number of genes (see Table S2). Both engineered strains showed greater differences in gene expression when compared with ATCC 10970 at the 24 h sampling time point (see Fig. 5). According to comparison of DNA content in OTC-production broths (Fig. S9; Supplementary Information 8), there were no consistent differences in growth rate between tested strains. Principal component analysis (PCA) plots displaying the biological variability of samples are presented in Fig. S8. To assess the normalization

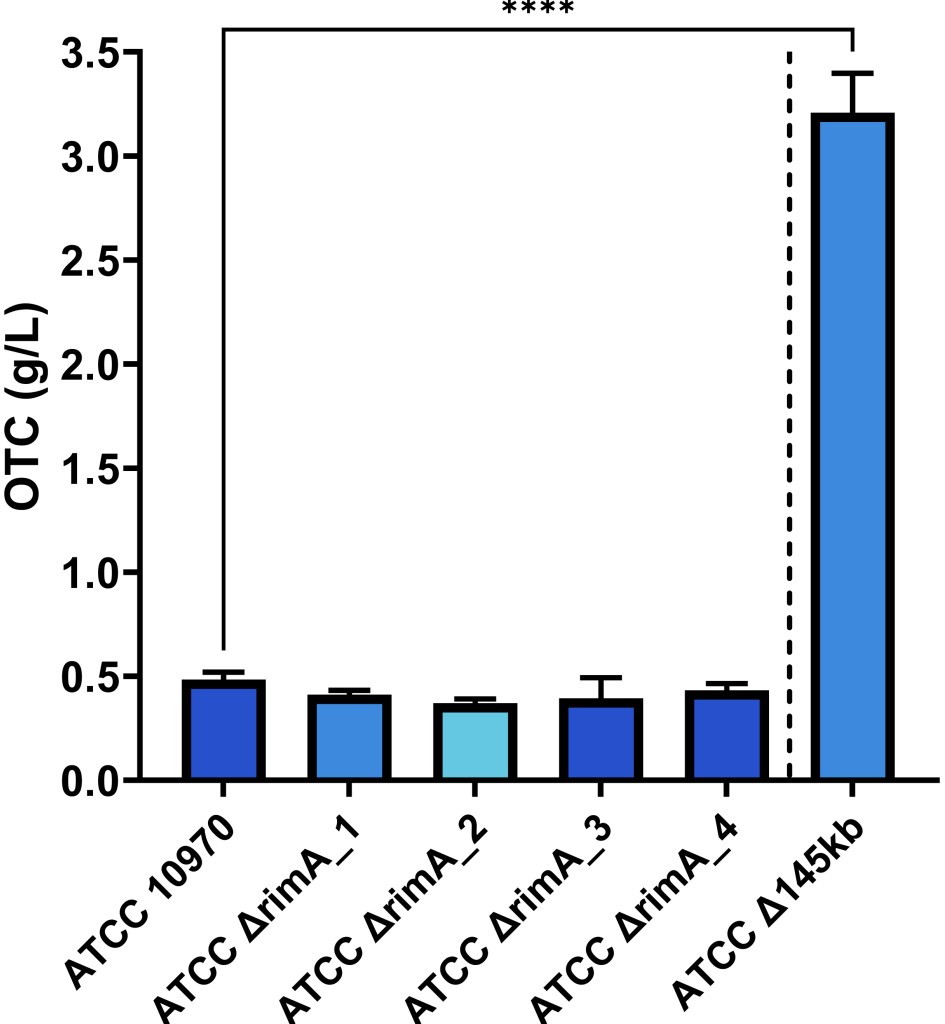

**FIG 4** The production of OTC by *S. rimosus* strains with inactivated rimocidin biosynthesis loading module (RimA) compared to control strains ATCC 10970 and ATCC Δ145 kb. Four independent engineered strains with inactivated *rimA* were tested (ΔrimA_1–ΔrimA_4). Mean is plotted with error bars showing s.d. ($n = 9$, three independent fermentations from three biological replicates). Significance is tested by Dunnett's T3-test, ****$P < 0.0001$.

process and evaluate the overall consistency of the data, we compared expression (TPM) of house-keeping genes *rpoB* in *gyrB* between all samples (33, 34) (Table S6). In addition, to verify the uniform growth phase of analyzed strains at both sampling time points, we compared TPM counts of *ftsZ* and *bldD* (Table S6), involved, respectively, in cell division and initiation of antibiotic production (35, 36). To gain deeper insight into biological processes altered in the ATCC Δ145kb and ATCC Δ240kb strains, we first performed gene ontology (GO) analysis (Supplementary information 7; Supplementary data 1). The most affected molecular function in the engineered strains was transmembrane transport (GO:0055085), with the largest proportion of both overexpressed and down-regulated genes. The ATCC Δ145kb strain also had changes in the regulation of DNA-templated transcription initiation (GO:0006352) and proteolysis (GO:0006508), which were not observed in the ATCC Δ240kb strain. Interestingly, ATCC Δ145kb strain also had a large number of affected genes related to metabolism of nitrogen compounds (GO:0006807). As *Streptomyces* genomes are not well annotated with GO terms, we suspected that GO analysis was not sufficient to cover all the differences between strains. Therefore, to gain a general perspective of observed changes in engineered strains and to emphasize the largest differences between engineered strains and native strain ATCC

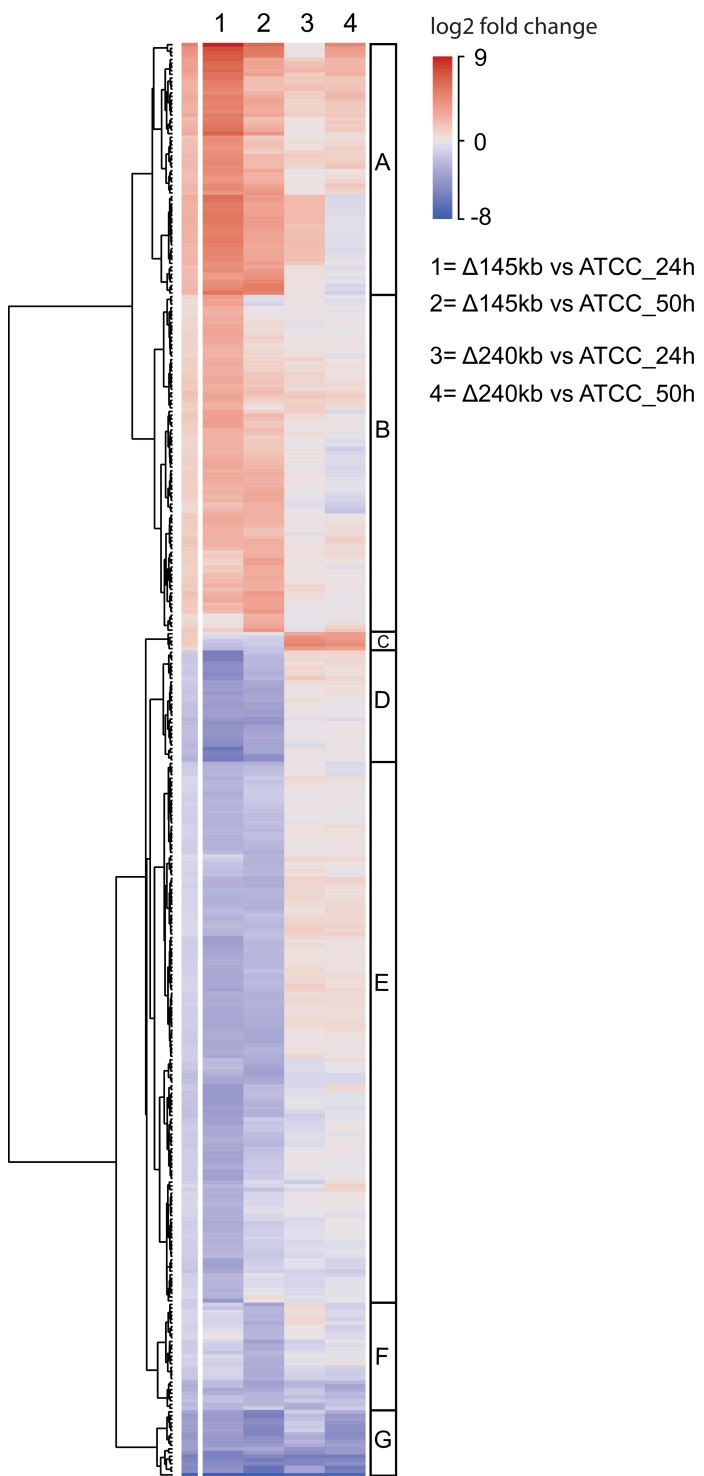

**FIG 5** Hierarchical clustering of genes with highly significant differential expression (log$^2$ fold change of >−2.5 or <2.5 in at least one engineered strain in at least one-time point—24 or 50 h). The heatmap is subdivided into seven clusters (A–G), based on observed expression dynamics between strains. Clusters C–G represent mostly downregulated genes, while clusters A and B consist of generally overexpressed genes.

10970, we performed a basic clustering analysis of RNA-sequencing data (see Materials and Methods). Clustering was performed on a data set of genes with log$^2$ fold change of >−2.5 or <2.5 in at least one engineered strain and at least one time point (24 or 50 h).

The results of clustering analysis are presented as a hierarchically clustered heatmap (Fig. 5).

Based on expression dynamics between analyzed strains, the heatmap was divided into seven clusters (A–G on Fig. 5). Clusters F and G represent mostly down-regulated genes, while clusters A and B represent genes that were generally overexpressed in both engineered strains. On the contrary clusters C, D and E were downregulated in the Δ145 kb strain but remain unaffected or are even up-regulated in the Δ240 kb deletion strain (see Fig. 5). These genes are probably responsible for observed differences between engineered strains (list of genes with annotations and expression levels—see Supplementary data 2). The most outstanding is cluster C (see Fig. 5), which is composed of only one operon with an unknown function (SRIM_017880–SRIM_017875) putatively encoding a Beta-lactamase family protein, UTRA domain-containing protein, ATP-binding protein, NAD(P)/FAD-dependent oxidoreductase, and one hypothetical protein.

Overall, genes involved in biosynthesis of secondary metabolites were the most affected group in both ATCC Δ145 kb and ATCC Δ240 kb strains. Almost exclusively, the genes in cluster G (Fig. 5) belong to secondary metabolism, namely, BGC 41 and BGC 46, which are strongly downregulated in both engineered strains. On the other hand, clusters A and B (C9 on Fig. 5) consist largely of genes from BGC 22 (biosynthesis of longicatenamycin B/C and yet unknown metabolite) and BGC 9—OTC, that were significantly overexpressed in both ATCC Δ145kb and ATCC Δ240kb strains. Together with genes involved in secondary metabolism, several proteases and ABC transporter proteins were also strongly upregulated as part of clusters A and B (see Supplementary Data 2). Overexpression of genes in clusters A and B is much more significant in the ATCC Δ145kb strain than the ATCC Δ240kb strain.

Interestingly, cluster D contains six putative genes involved in the synthesis of the rodlet layer, a typical surface layer of both aerial hyphae and spores in *Streptomyces*. Two rodlin genes *rdlA* (37) and four different chaplin (38) genes are significantly down-regulated only in the Δ145 kb strain. In *Streptomyces* species, the chaplin genes are, to some extent, redundant, but *rdlA* is indispensable for the establishment of this rodlet structure (37). Also, four other chaplin and rodlin genes were identified as downregulated in the Δ145 kb strain as part of cluster E (Fig. 5). This suggests that changes in the rodlet layer of ATCC Δ145kb might explain the morphological changes in this strain (Fig. S6). The large cluster E consists of several poorly annotated or hypothetical genes (presented in Supplementary data 2). We also performed a separate analysis focusing on the number of differentially expressed genes in 100 kb intervals across the genomes of engineered strains (Fig. S10). Interestingly, genes from the OTC BGC and BGC 22 are not only the most highly over-expressed genes (Fig. 6) but are also the two largest intervals of differentially expressed genes. Importantly, the greatest transcriptome changes were observed in identical regions for all engineered strains (Fig. S10) but are most pronounced in the ATCC Δ145kb strain.

## Comparative bioinformatics analysis of BGC expression in ATCC 10970 and engineered strains

RNA-seq data demonstrated that increases in OTC biosynthesis of the engineered strains are correlated with a significant increase in expression of OTC biosynthetic genes, which were the most overexpressed genes in both ATCC Δ145kb and ATCC Δ240kb strains (Fig. 6), with average $\log^2$ fold ratios of 4.7 and 1.7, respectively (Table 1).

Alongside BGC 9 (OTC BGC), a cryptic cluster annotated as BGC 22 caught our attention due to over-expression (>32-fold change, Table 1; Fig. 6) in the ATCC Δ145 kb strain. Upon more detailed analysis, we propose that the BGC 22 region consists of two separate BGCs each with its own NRPS and accessory genes, annotated here as 22_1 (SRIM_030975–SRIM_031065) and 22_2 (SRIM_030895–SRIM_030970). The two clusters are adjacent, possibly share accessory genes, and are overexpressed with identical intensity (see Fig. S2.1). The first part (BGC 22_1) encodes a cyclic hexapeptide longicate-namycin B, recently discovered in *S. rimosus* by Li et al. (39), and the second part (BGC

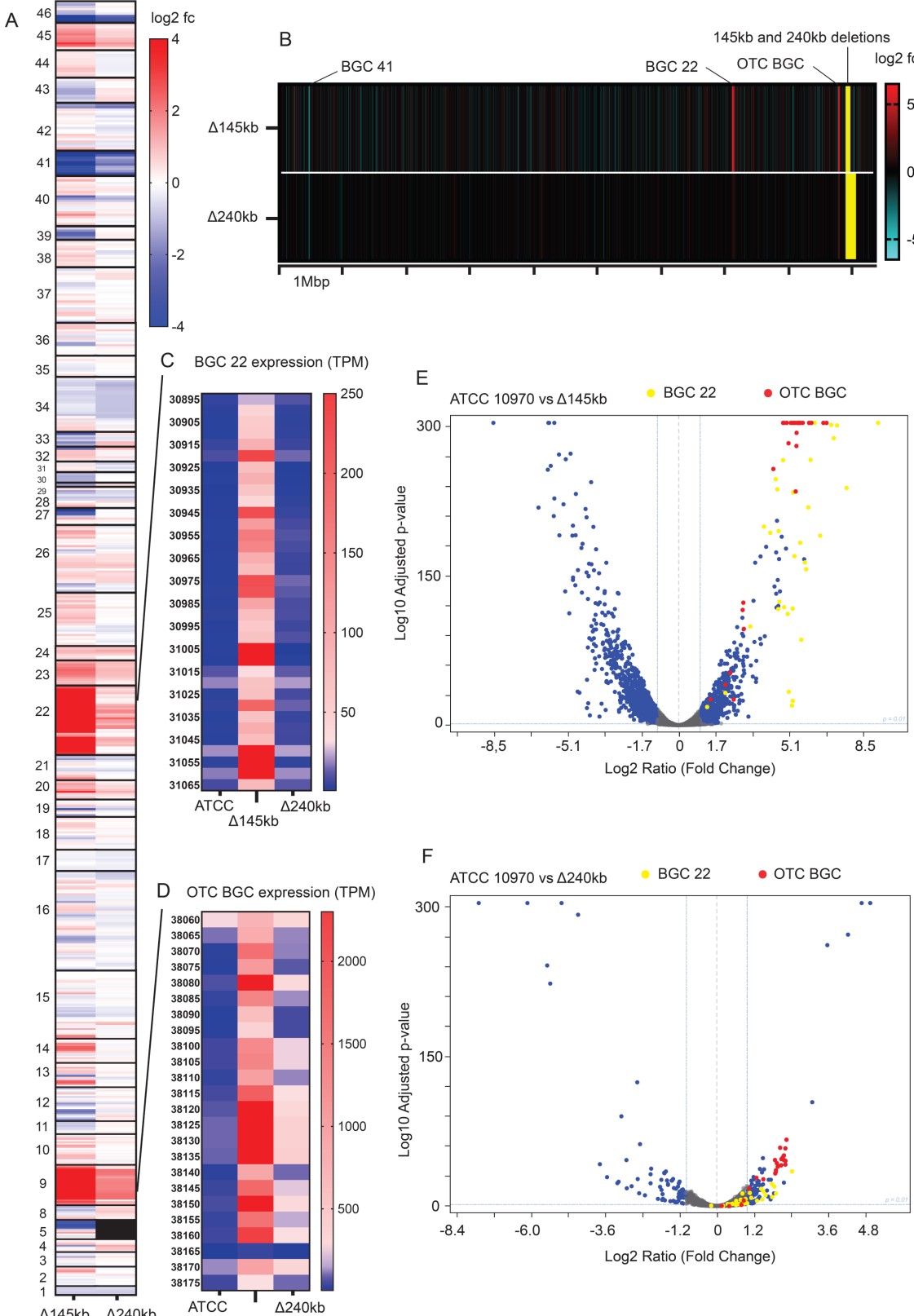

**FIG 6** Comparative transcriptome analysis of *S. rimosus* ATCC 10970 and engineered strains ATCC Δ145kb and ATCC Δ240kb at the first sampling time point (24 h). (A) Transcriptome heatmap displaying relative expression (log² fold change) of BGCs in genome-reduced strains compared to control ATCC 10970. (B) Transcriptional background of the genome reduced strains, changes in relative expression (log² fold change) compared to ATCC 10970. (C) Heatmap

**FIG 6** (Continued)

displaying expression values of entire BGC22 in four tested strains (TPM-transcripts per million; SRIM_ gene identifiers are presented). (D) Heatmap displaying expression values of entire OTC cluster (BGC 9) in four tested strains. (E and F) Volcano Plot depicting transcriptome changes when comparing ATCC Δ145 kb (E) and ATCC Δ240 kb (F) to control ATCC 10970. The $x$-axis represents the $\log^2$ fold change in gene expression, and the $y$-axis represents the statistical significance as −log10 adjusted $P$-value. Each data point represents an individual gene, and significant differentially expressed genes are highlighted ($\log^2$ ratio >1, $\log^2$ ratio <−1, log10 $P$ value > 2).

22_2) encodes a yet unknown NRPS metabolite (AntiSMASH analysis—Fig. S2). The entire BGC 22 was moderately overexpressed in the ATCC Δ240kb strain (1.86-fold change) (Table 1; Fig. 6F).

Despite the observed overexpression of BGC 22 in ATCC Δ145 kb, the absolute expression strength of cluster BGC 22 was only around 10% in comparison to the OTC BGC (see TPM scale at Fig. 6A and B). Therefore, the magnitude of BGC 22 overexpression is so high, considering it originates from its almost cryptic level of expression in ATCC 10970 (Fig. 6C). Importantly, BGC 22_1 encoding longicatenamycin congeners has been considered cryptic in *S. rimosus*, and until now its activation could only be detected when using high-throughput elicitor screening (39). Similarly, longicatenamides A–D were detected in *Streptomyces* cultures only when using a co-cultivation approach (40). Our results, therefore, indicate that targeted genome reductions can be used to induce expression of cryptic clusters. Also, it is important to emphasize that the expression of other BGCs, of which many are un-characterized, was also significantly affected in the ATCC Δ145 kb and ATCC Δ240 kb strains (see Table 1). A Volcano Plot depicting transcriptome changes in a second independent strain with 145 kb deletion (ATCC Δ145 kb-b) is presented in Fig. S4B. Importantly, OTC BGC and BGC 22 remain the most overexpressed groups of genes in the ATCC Δ145 kb-b strain, confirming the effect of the 145 kb deletion on observed BGC overexpression.

## Comparative analysis of metabolite profiles of *S. rimosus* ATCC 10970 and engineered strains by applying MS analysis

Full-scan MS analysis of the engineered strains was done to assess the impact of the two deletions on metabolite production profiles. We immediately observed peaks belonging to oxytetracycline (OTC; 461,7 [M + H]⁺obs/460,4340 [M]) (OTC on Fig. 7) and rimocidins in *S. rimosus* ATCC 10970 (RIM on Fig. 7A). On full-scan LC-MS chromatograms (Fig. S11), we could distinguish between production of five rimocidin congeners in the ATCC 10970 strain background. Additionally, in the next step, using high-resolution mass spectrometry (HR-MS), we confirmed the production of rimocidin, CE-108, rimocidin (27-etyl), CE-108B, and rimocidin B in the ATCC 10970 background (HR-MS data—Table S7, RIM structures—Fig. S13). The ATCC 10970 ΔOTC strain was included in the metabolite profile analysis to investigate if deletion of the OTC BGC affects production of other metabolites. Despite the removal of OTC as the most abundant metabolite, the metabolome profile

**TABLE 1** Significantly affected[a] BGCs after genome reductions in engineered *S. rimosus* strains[b]

| Significantly affected BGC | Metabolite | Fold change-$\log^2$ ratio ($x$ vs ATCC 10970) | |
| --- | --- | --- | --- |
| | | ATCC Δ145 kb | ATCC Δ240 kb |
| BGC 22_1 | Longicatenamycin B | 5.36 | 0.88 |
| BGC 22_2 | Unknown | 5.15 | 0.91 |
| BGC 9—OTC | Oxytetracycline | 4.7 | 1.7 |
| BGC 45 | Momomycin | 1.73 | 0.79 |
| BGC 23 | Tyrobetaine-2 | 1.74 | 0.85 |
| BGC 46 | Unknown | −1.97 | −1.02 |
| BGC 5 | Unknown | −2.33 | / |
| BGC 41 | Unknown | −4.09 | −2.33 |

[a]BGCs with an average gene $\log^2$ fold change of >1 or <−1 in at least one mutant strain was considered significantly affected.
[b]Average $\log^2$ ratio—fold change of expression at 24h time point.

of the ATCC 10970 ΔOTC strain did not change, with the exception of a minor increase in base peak intensity belonging to the rimocidins (see Fig. 7A, *Y* axis on Fig. S11). Interestingly, the precise *in frame* deletion of *rimA*, encoding the polyketide synthase loading module of rimocidin BGC did not have any effect on OTC production (see Fig. 9C). As expected, no rimocidin congeners could be detected in the Δ145 kb and Δ240 kb deletion mutants (see Fig. 7B, C, E and F and Fig. S11). Comparable to the HPLC results (Fig. 3), differences in OTC titers between ATCC 10970 and ATCC Δ145 kb and ATCC Δ240 kb strains can clearly be observed also on full-scan-LC-MS spectra (see Fig. 7A through C). The absence of the OTC BGC accompanied with the deletion of 145 kb region in ATCC 10970 ΔOTC Δ145 kb strain resulted in a clean background (Fig. 8); the metabolic profile contained virtually no major metabolites.

In addition to OTC and rimocidin congeners, we also attempted to identify metabolites related to cryptic BGCs, which were noticed through transcriptomics analysis (Table 1; Fig. 6A). Accordingly, full-scan LC-MS analysis of the ATCC Δ145 kb sample revealed three newly emerged peaks that became even more evident when metabolites were extracted using 4 volumes of acetonitrile (ACN) ($[M + H]^+_{obs}$ 387.7; 764.9; and 778.9 (peaks 1, 2, and 3 on Fig. 9; see also Table 2). Peak no. 1 was predicted to be tyrobetaine-2, a member of the tyrobetaines, a class of nonribosomal peptides with an unusual trimethylammonium tyrosine residue (42). Tyrobetaine-2 was also detected in the control ATCC 10970 extract, but with lower abundance (*Y* axis on Fig. S12A). Interestingly, the tyrobetaine BGC (BGC 23) was significantly overexpressed in the Δ145 kb strain (Fig. 6A; Table 1), which is consistent with the LC-MS results (Fig. 9B).

The other two new peaks ($[M + H]^+_{obs}$ 764.9 and 778.9) were considered to be potential cryptic metabolites produced by one of the overexpressed BGC in the ATCC Δ145 kb strain (Table 2). Indeed, peak no. 2 was identified as longycatenamycin B (39, 43) in the next step using HR-MS ($[M + H]^+$obs 763.3546, $[M + H]^+$cal 763.3546). BGC 22_1, encoding longycatenamycin B, and BGC 22_2 are the most overexpressed BGCs in the engineered strains, which is consistent with the appearance of this metabolite.

We speculated that peak no. 3 (Fig. 9B) represents a new metabolite produced by uncharacterized NRPS complex BGC 22_2 (presented in Fig. S2 and S2.1), considering its strong overexpression in the ATCC Δ145 kb strain. To our surprise, the HR-MS results (Table S7) indicated that peak no. 3 is, in fact, another member of the longicatenamycin (S-520 antibiotic) family, synthesized by the same NRPS as longicatenamycin B (BGC No. 22_1), with an additional methyl group (D-Val→Ile) ($[M + H]^+_{obs}$ 777.3699, $[M + H]^+_{cal}$ 777.3702). This longycatenamycin congener was detected before by von Nussbaum et al. (43) and possibly by Shoji and Sakazaki (44), but only as a component of a complex mixture of congeners and without a specific name. We therefore designated this congener as longicatenamycin C (Table S7).

No longicatenamycin could be detected by LC-MS in the native ATCC 10970 strain. We also attempted to evaluate if the presence of longicatenamycins in the ATCC 10970 strain could be masked by RIM congeners, which elute at same retention time and have similar $[M + H]^+_{obs}$ (see Fig. S7). We therefore analyzed chromatograms obtained by full-scan MS of the *rimA* mutant strain, not producing RIM and found traces of both longicatenamycin B and C. In accordance with the work by Li et al. (39) where longicatenamycin B was first characterized, these authors also used a RIM non-producing strain of *S. rimosus*. To conclude, we show here that longicatenamycins B and C are present in trace amounts in the ATCC 10970 parent strain, masked to LC-MS detection by larger amounts of RIM congeners.

## Repositioning of the entire OTC BGC in *S. rimosus* ATCC Δ145 kb and ATCC Δ240 kb strains

To explore the effect on OTC titer by the location of the OTC BGC and gene dosage, we designed an OTC cluster relocation experiment similar to that described before (29). We introduced the identical 145 kb deletion into the genome of the *S. rimosus* ATCC 10970 ΔOTC strain, which has almost the entire OTC BGC deleted and, therefore, does not

produce oxytetracycline (see Supplementary information 3 and 8). Three independent *S. rimosus* ATCC 10970 ΔOTC Δ145 kb colonies were then complemented with a pYAC-ΦC31-Ts-OTC construct containing the entire OTC BGC integrated at the ΦC31 *attB* site. This results in the re-location of the OTC BGC from the left terminal arm to the *attB* site—located in the central region of *S. rimosus* genome (29) (Fig. 10C). Similarly, duplication of

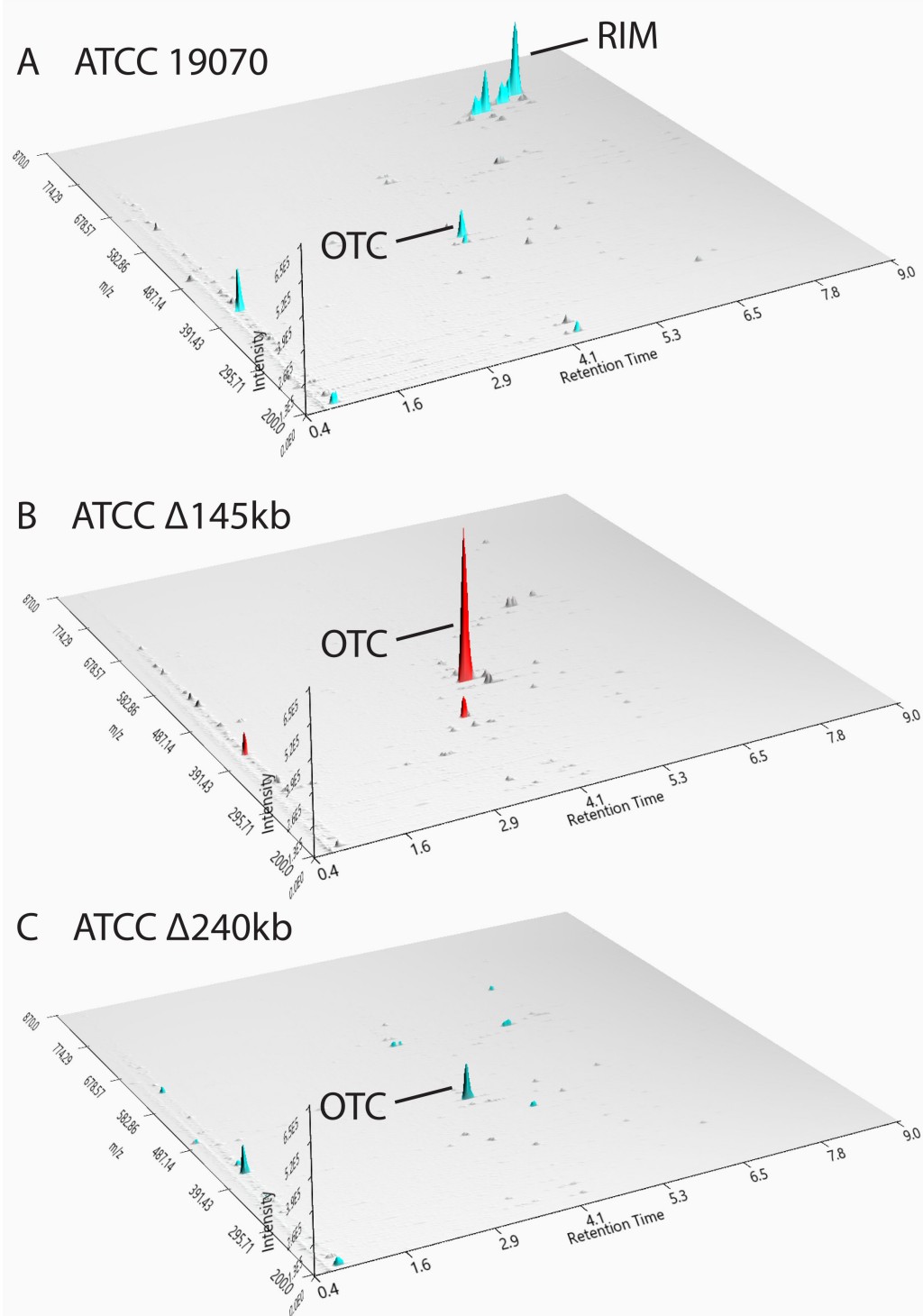

**FIG 7** Full-scan LC-MS analysis of metabolite profiles of ATCC 10970 (A) and engineered strains ATCC Δ145 kb (B) and ATCC Δ240 kb (C). RIM—rimocidins, OTC—oxytetracycline. [M + H]$^+$obs—see Table 2. Visualization with MZmine 3.0 (41).

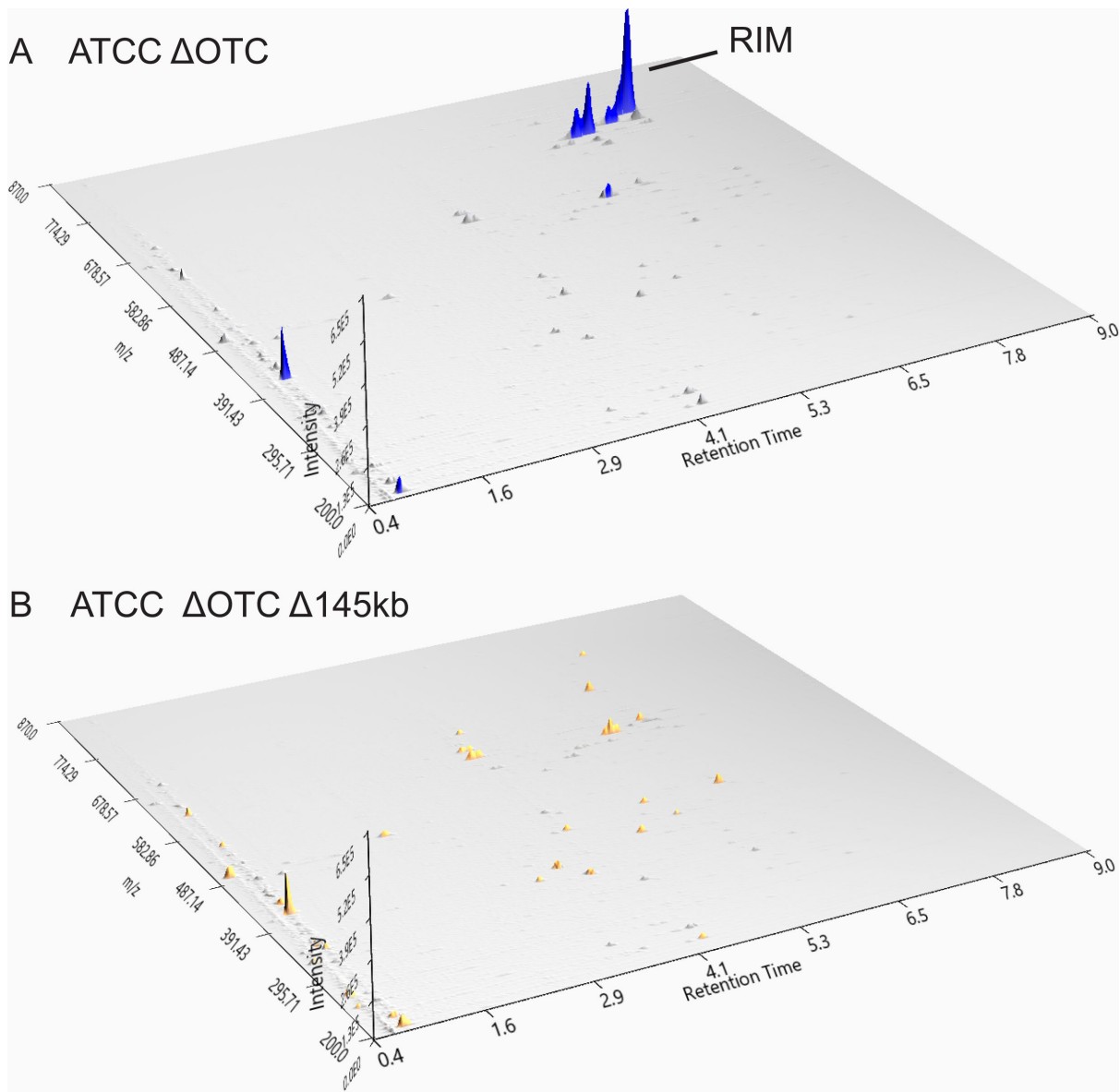

**FIG 8** Full-scan LC-MS analysis of OTC-depleted engineered strains engineered strains ATCC ΔOTC (A) and ATCC ΔOTC Δ145 kb (B). RIM—rimocidins, OTC—oxytetracycline. [M + H]$^{+}$obs—see Table 2. Visualization with MZmine 3.0 (41).

the OTC BGC was achieved by introducing pYAC-ΦC31-Ts-OTC plasmid into the *attB* site of the ATCC Δ145 kb strain (see Fig. 10B).

### Evaluation of the OTC titer and morphological properties of the engineered strains *S. rimosus ATCC10970ΔOTCΔ145 kb::OTC and S. rimosus ATCC10970Δ145 kb::OTC*

All three isolates of ATCC 10970 ΔOTC Δ145 kb::OTC had increased OTC production with an average OTC titer of 3 g/L (Fig. 11A). Pikl et al. (29) had already demonstrated that relocation of the OTC cluster to a more central region of the chromosome of the ATCC 10970 parent strain had a positive effect on OTC production, resulting in an OTC titer increase from around 200 mg/L to around 1 g/L.

However, to evaluate the effect of copy number of the OTC BGC on OTC titer, we also introduced the pYAC-ΦC31-Ts-OTC plasmid (29) containing the entire OTC BGC into the ATCC Δ145 kb strain, thus generating a strain containing a second copy of the OTC BGC

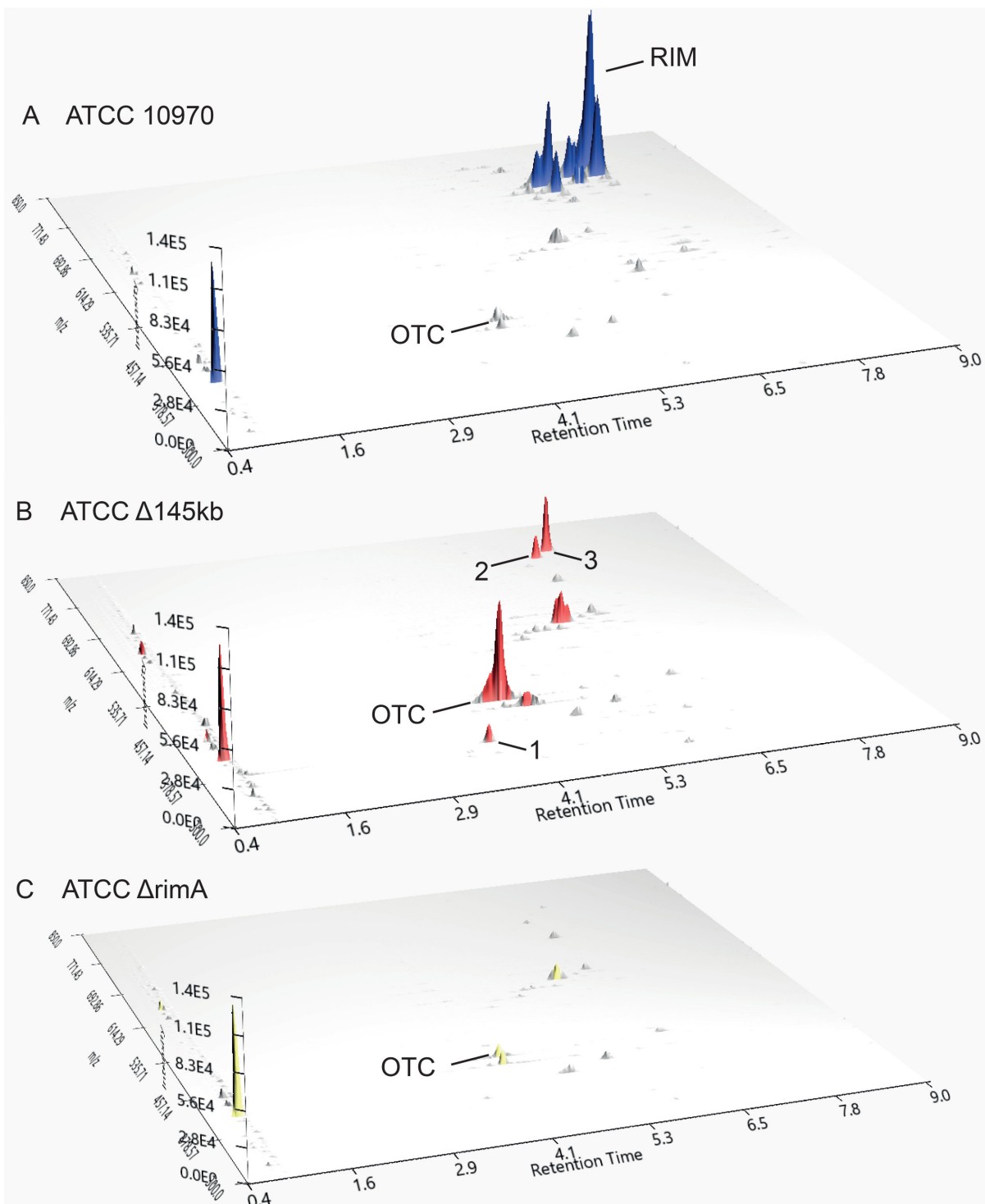

FIG 9 Full-scan LC-MS analysis of *S. rimosus* ATCC 10970 (A), ATCC 10970 Δ145 kb (B) and ATCC 10970 Δ*rimA* production broths after acetonitrile extraction. Emergence of three new metabolites in the strain ATCC Δ145kb is visible on chromatogram B. RIM—rimocidins, OTC—oxytetracycline ([M + H]$^+_{obs}$—see Table 2), 1—tyrobetaine-2 ([M + H]$^+_{obs}$ = 387.7), 2—longicatenamycin B ([M + H]$^+_{obs}$ = 764.9), 3—longicatenamycin C ([M + H]$^+_{obs}$ = 778.9). Compete removal of all rimocidin-related metabolites can be observed for *rimA* mutant, while OTC titer remain unaffected compared to ATCC 10970. Visualization with MZmine 3.0 (41).

in the ATCC Δ145 kb + φC31 OTC strain (see Fig. 10B). Remarkably, the transformant (ATCC Δ145 kb + pYAC-ΦC31-Ts-OTC) had a titer of 5.2 g/L of OTC (Fig. 11), which is almost 10× higher compared to the parent ATCC 10970 strain and almost double the

**TABLE 2** Major peaks/metabolites detected on full-scan LC-MS analysis in engineered strains and ATCC 10970 parent strain[a]

| BGC no. | Metabolite | Base peak intensity in strain | | | $[M + H]^{+}$obs/[M] |
| --- | --- | --- | --- | --- | --- |
| | | Δ145 kb | Δ240 kb | ATCC 10970 | |
| 9 | Oxytetracycline | +++ | ++ | + | 461.7/460.4 |
| 6 | Rimocidins | – | – | +++[b] | 739.4; 740.3; 754.3; 768.34/[b] |
| 22_1 | Longicatenamycin B | ++ | + | – | 764.8/763.3[b] |
| 22_2 | Longicatenamycin C | ++ | + | – | 778.9[b]/777.4[b] |
| 23 | Tyrobetaine-2 | ++ | + | + | 387.7/387.4 |

[a]+++, Base peak intensity >5.0 E[5]; ++, base peak intensity >1.0 E[5]; +, base peak intensity >5.0E[3]; –, not detected.
[b]See also HR-MS data (Table S7).

amount (2.8 g/L, Fig. 2) produced by the ATCC Δ145 kb strain. Considering the titers of industrial strains M4018 and R6-500, we have demonstrated that industry-relevant OTC titers can be achieved by the introduction of a second copy of the OTC BGC to the genome of native parent strain ATCC 10970 containing the 145 kb deletion (Fig. 11 compared to Fig. 2).

### Limited re-optimization of the cultivation medium

Naturally, medium composition and cultivation conditions have profound effects on any bioprocess and final titer of the target product (OTC in our case). However, to be able to compare different engineered strains of *S. rimosus* carried out at the laboratory (shaker) scale, we routinely use GOTC-P medium to evaluate OTC production capacity.

However, production medium GOTC-P only contains limited amounts of carbon and nitrogen sources when compared with the industrial media. We therefore, set very limited medium optimization experiment, where we only varied the content and ratios of carbon and nitrogen sources (Fig. S14). Interestingly, the increase of soy-flour did not result in the increase of the OTC titer (data not presented). In contrast, the increase of the carbon source (corn starch) had a profound influence on the titer of OTC. Surprisingly

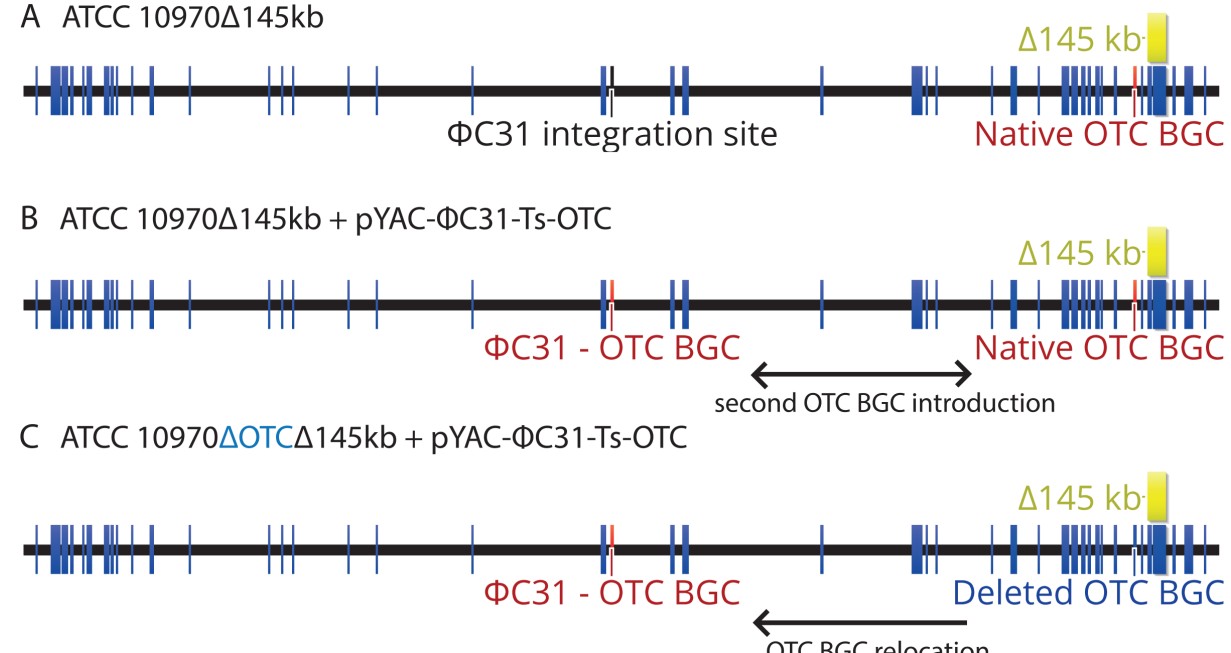

**FIG 10** Re-introduction of the entire OTC gene cluster in the chromosome of *S. rimosus* ATCC 10970 Δ145 kb (A). Introduction of the pYAC-ΦC31-Ts-OTC vector into *S. rimosus* ATCC 10970Δ145 kb, resulting in OTC BGC duplication (B) and OTC gene cluster re-location (C) when introduced to *S. rimosus* ATCC 10970 ΔOTC Δ145 kb strain background.

however, the increase of the carbon source had more profound effect on the biosynthesis of OTC in strain ATCC Δ145 kb + φC31OTC containing additional copy of OTC BGC in addition to the 145 kb deletion. Increase of the carbon source had limited influence on the titer of OTC (around 13%) with the ATCC Δ145 kb strain. To conclude, an increase in carbon source by 15% resulted in a significant increase (by 30%) in the final titer of OTC in the ATCC Δ145 kb + φC31OTC for additional 30%.

## DISCUSSION

Pan-genome and functional genome analyses of *Streptomyces* chromosomes have shown that most conserved genes responsible for the fundamental functions of cell viability are concentrated in the core (central) region of the *Streptomyces* chromosome (28, 45). By contrast, the vast majority of BGC encoding biosynthesis of diverse secondary metabolites are located in the sub-telomeric regions. It has been demonstrated that the terminal diversity of chromosomes of *Streptomyces* species correlates with intense DNA plasticity (46) which can include the occurrence of large deletions (often accompanied by large DNA fragment amplifications) and more complex genome rearrangements that can be associated with circularization and chromosomal arm exchange (47–50). Interestingly, Zhang et al. (51) identified a sub-population of cells that arose spontaneously in *Streptomyces coelicolor* and hyperproduced the antibiotic actinorhodin. This sub-population of mutant cells declines in fitness, while losing fragments of chromosomal ends. Differentiation into this hyperproducing phenotype was, therefore, accompanied by huge loss of fitness due to massive deletions of up to 1 Mb. However, the instability of the sub-telomeric region is not necessarily lethal to the cell, as observed with several *Streptomyces* species (51–53).

As exemplified by *S. rimosus* and other *Streptomyces* species, both the frequency and scale of rearrangements can also be induced by mutagenesis or genetic engineering (50, 54). Jo et al. (55) demonstrated that a rapamycin-overproducing strain of *Streptomyces rapamycinicus*, generated by random mutagenesis and strain selection for higher production of rapamycin, underwent spontaneous genome reduction: the 12.47 Mb genome of the native parent strain was reduced to 9.56 Mb in the overproducing strain. A larger gene deletion near one chromosomal end was accompanied by a DNA duplication, containing the entire BGC encoding biosynthesis of rapamycin. Therefore, it can be concluded that large-scale deletions could be carried out in sub-telomeric regions, hence, still avoiding significant disabilities to the viability of engineered *Streptomyces* strains. We have demonstrated that in the *S. rimosus* industrial strains M4018 and R6-500, which were developed by random mutagenesis over decades of intensive strain improvement regimes aimed at high OTC titer (19, 24, 56), DNA deletions and rearrangements occurred in sub-telomeric regions of the linear chromosome. We therefore, hypothesized that these gene rearrangements may have influenced biosynthesis of OTC, considering that the selection process was based on OTC titer. We thus, recognized that these two strains could serve as exemplars, on which we could design targeted genome-engineering experiments in the *S. rimosus* ATCC 10970 parent strain with the aim of modulation of biosynthesis of OTC.

However, several questions arise when considering genome reduction, such as the choice of chromosomal region that should be reduced and the size of the reduction that should be introduced at that chromosomal location (57, 58). A few attempts to reduce genomes in *Streptomyces* species are described in the literature (59–61). For example, in a comprehensive effort, Zhuo et al. (60) reported sequential deletion of 10 PKS and NRPS clusters in *S. coelicolor* M145 including a 900 kb deletion of the sub-telomeric region. In total, 1.22 Mb of the chromosome was removed (14% of the genome). An artificially circularized genome of *S. coelicolor* M145 was also reported (59), including the deletion of 1.6 Mb of genome: 840 kb from the left and 761 kb from the right arm of the linear genome of *S. coelicolor*. No obvious differences in growth rate or strain morphology were observed. Interestingly, deletion of the 900 kb telomeric region resulted in a decrease in actinorhodin titer compared to the *S. coelicolor* M145 parent strain, suggesting that some

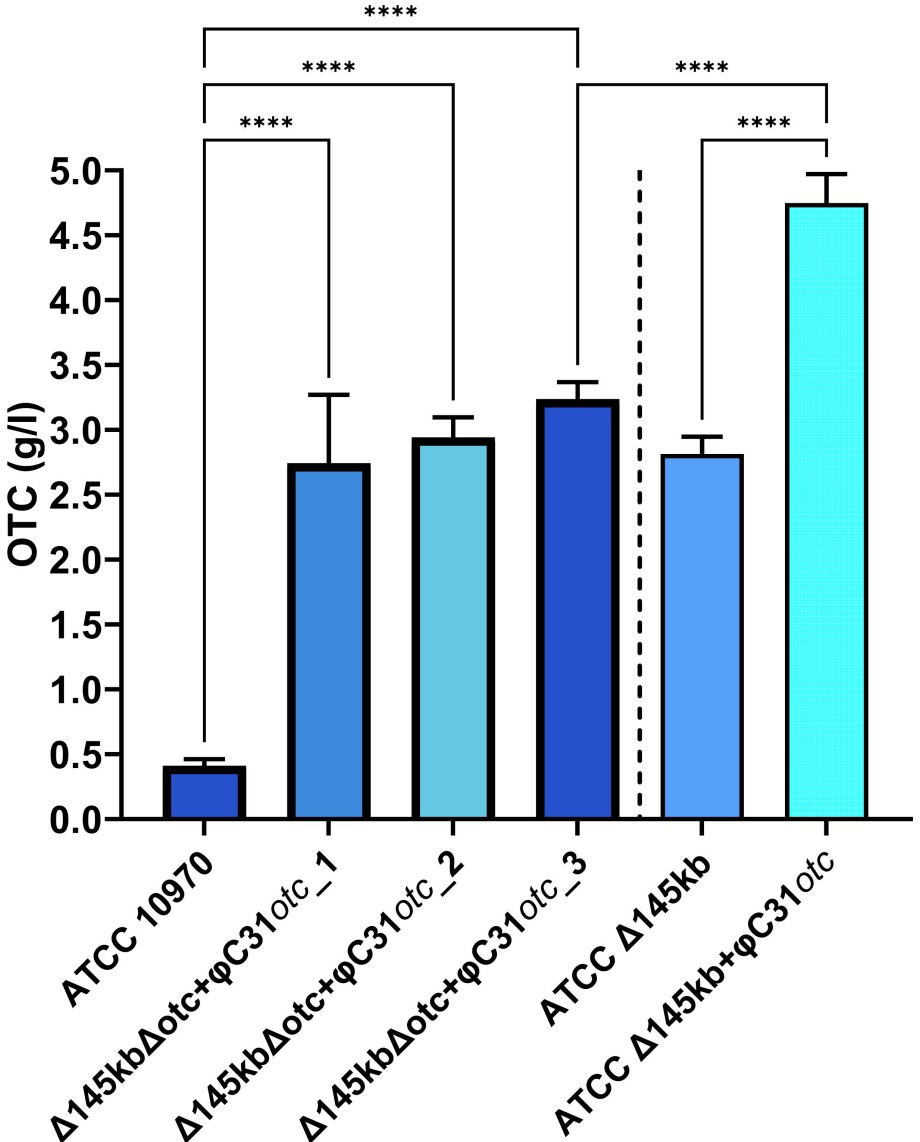

**FIG 11** Engineered strains *S. rimosus* ATCC10970ΔOTCΔ145 kb::OTC with re-located OTC BGC (A) and two *S. rimosus* ATCC10970Δ145 kb strains containing a second copy of the OTC BGC (B). Mean is plotted with error bars showing s.d. (*n* = 9, three independent fermentations from three biological replicates). Significance is tested by Dunnett's T3-test, ****$P$ < 0.0001.

unknown genes from the 900 kb sub-telomeric region affect expression of the actinorhodin BGC (58). In another attempt, Komatsu et al. (62) reported a stepwise deletion of sub-telomeric regions of *Streptomyces avermitilis*, an important industrial producer of the anthelmintic agent, avermectin. The most redundant strain, SUKA17, had a deletion spanning 1.67 Mb (18.5% of the genome) into the sub-telomeric region. This strain was able to grow on minimal medium; therefore, the extensive chromosome deletion was neither lethal not influential on primary metabolism. To summarize, the production of target metabolites by these genome-reduced strains did not result in an industrially relevant titer increase. After identifying the genome-rearrangements in the assembled genomes of industrial strains M4018 and R6-500, we were then particularly interested in evaluating the effect of specific DNA deletions on the terminal parts of the *S. rimosus* genome. As described in the introduction section, we therefore introduced simple and precise deletions in the vicinity of the OTC BGC in the ATCC 10970 parental strain

background. We generated deletions of 145 and 240 kb by the application of an in-lab optimized CRISPR-Cas9 tool (26). Remarkably, the deletion of a 145 kb region of the *S. rimosus* ATCC 10970 parent strain, located in the vicinity of the OTC BGC, had a profound influence on the OTC titer—immediately achieving high levels of OTC production in this one-step targeted modification that was inspired by shared differences of both M4018 and R6-500 used for industrial production of OTC to the parental ATCC 10970 wild-type strain. We also observed and rationalized the change in morphology of the strain with the 145 kb deletion. Importantly, the deletion of the 145 kb region in the vicinity of the OTC BGC in the ATCC 10970 strain not only resulted in an overall increase in OTC titer, but also intensive production of OTC was initiated within 24 h of commencement of the fermentation. By contrast, OTC biosynthesis was only initiated after 48 h of fermentation by the ATCC 10970 strain.

The transcriptome data fully support the hypothesis that the main reason for such an increase in OTC biosynthesis was due to enhanced transcription of the OTC BGC. Introduction of a second copy of the OTC BGC by chromosomal integration resulted in a further significant increase in OTC production, thus supporting the observation that increasing gene dosage contributes significantly to the OTC titer increase. It is likely that by further increasing the gene dosage (integration of additional copies of OTC BGC), a further increase in OTC titer could be expected.

What is the mechanism that causes such a profound increase in transcription of OTC biosynthetic genes in the strain with the deletion of 145 kb in the vicinity of OTC BGC? A large proportion of the coding regions present in this DNA fragment is the BGC encoding the polyene antibiotic rimocidin (RIM). It is possible that the deletion of the RIM BGC could result in this phenotype, as reported in the literature (31). This could be due to an unknown RIM-OTC BGCs cross-regulatory effect, or perhaps simply an increase in availability of the malonyl-CoA substrate for OTC biosynthesis, considering that malonyl-CoA is the building block for biosynthesis of both antibiotics, OTC and RIM. At this point, it is not possible to make any definite conclusion about a potential RIM-OTC BGCs cross-regulatory effect. However, considering that the ATCC Δ240kb strain contains a much larger deletion including the RIM BGC, and we did not observe such a strong OTC overproduction phenotype, this is unlikely. However, to exclude this possibility and to evaluate the potential effect of inactivation of the RIM BGC, we carried out a precise deletion of the loading domain of the RIM PKS in the ATCC 10970 parental strain, thus disabling the functionality of the RIM PKS, but preserving most of the RIM BGC. No increase in OTC titer was observed. Therefore, we concluded that deletion of the RIM BGC is not the likely cause of the increase in OTC titer.

Numerous research efforts have shown (12, 15, 63) that mutations in regulatory genes, genes encoding sigma factors, resistance genes, and export proteins could have significant effects on secondary metabolite production. We identified only a handful of other genes (not belonging to the RIM BGC) which were deleted in strain ATCC Δ145 kb. For example, one putative gene with homology to an unknown helix-turn-helix transcriptional regulator and two putative genes belonging to the RNA polymerase sigma-70 factor family were identified within the 145 kb region. However, based on the proposed functions of these genes (Table S2), it is difficult to draw any conclusion on their potential regulatory role in OTC biosynthesis in the ATCC Δ145kb strain. However, as these genes were also deleted in the ATCC Δ240kb strain (with modest increase in OTC titer), it is unlikely that they are major contributors to the OTC overproducing phenotype.

The OTC BGC contains 24 genes, including those involved in the regulation of OTC biosynthesis. Overexpression of the regulator genes *otcR*, *otcG,* and *oxyTA1*, although not industrially significant, causes an increase in OTC titers (19). However, no mutations were identified in the regulatory genes present in the OTC BGC in M4018 and R6-500 when compared to ATCC 10970. In fact, the entire DNA sequence of the OTC BGC is identical in these three strains. Based on our experiments, we conclude that pathway-specific regulatory genes present in OTC BGC are not the driving force behind the OTC titer increase seen in the industrial strains.

We have analyzed in detail two strains with 145 kb deletion: ATCC Δ145 kb(-a) and ATCC Δ145 kb-b. The OTC titer achieved by these two strains is significantly higher and very similar RNA-seq expression patterns were observed in these two strains. However, in ATCC Δ145kb(-a), the titer of OTC is superior and OTC BGC gene expression is also further enhanced in this strain. ATCC Δ145kb(-a) and ATCC Δ145kb-b are two independently generated strains. Each has a few additional mutations, specific for each strain (Table S5). This variation may be due to sequence changes through spontaneous mutation. The superior performance of the ATCC Δ145kb(-a) strain, compared to ATCC Δ145kb-b, may potentially be due to unspecified mutations (Table S5). However, at this point, it is impossible to give any more specific comment on this.

We observed this slight reduction in pH value by the ATCC Δ145kb(-a) strain during fermentation. The content of biomass is very similar between different strains, and RNA-seq analysis did not highlight any significant change in gene expression patterns of primary metabolic pathways. The slightly lower pH value of the ATCC Δ145kb(-a) strain during fermentation might be due to faster consumption of simple sugars from soy-flour, or faster degradation of starch, resulting in slightly more intensive carbon flux through glycolysis or the TCA cycle and consequential production of organic acids. As discussed earlier, telomeres of linear chromosomes of *Streptomyces* species contain long-terminal inverted repeats and covalently bound terminal proteins (TPs). Recent studies have shown that, in *Streptomyces* species, substantial chromosomal remodeling occurs during the life cycle, with the chromosome undergoing rearrangements from an "open" to a "closed" conformation (28, 64). Szafran et al. (65) showed that, in *S. venezuelae*, the arms of the sub-telomeric regions were separated spatially at entry to sporulation, but during sporogenic cell division, the process was often accompanied by secondary metabolite biosynthesis, which was aligned closely with the core of the chromosome. The question arises, whether a 145 kb deletion in the sub-telomeric region and in the vicinity of the OTC BGC could cause such a change in the topology of the chromosome to trigger an increase in OTC BGC gene expression. A recent observation by Deng et al. (66), who studied 3D organization of the chromosome in the model strain *S. coelicolor* showed that the chromosome of *S. coelicolor* undergoes structural changes from primary to secondary metabolism, while the DNA encoding some BGCs forms special local structures of chromatin when highly expressed. These authors also identified specific regions of the *S. coelicolor* chromosome, which enabled very high expression of externally integrated reporter genes or even entire BGCs, thus demonstrating that certain chromosomal locations significantly affect the expression levels of the local genes, whether endogenous or exogenous. A similar observation was reported by Pikl et al. (29) in *Streptomyces rimosus*, where relocation of the entire OTC gene cluster to an alternative position on the chromosome resulted in significant increase in titer of OTC. Therefore, location of the BGC on the chromosome seems to play an important role in its gene expression.

The study by Deng et al. (66) and other studies on the topology of *Streptomyces* chromosomes (64, 65) also suggest that 3D structure and potential interactions of different chromosomal locations may play a very profound role on the biosynthesis of secondary metabolites. Considering our experimental results, it is quite possible that the deletion introduced in the vicinity of the OTC BGC caused changes in chromosome topology, hence increased expression of genes located in the OTC BGC.

Although in our work we primarily targeted biosynthesis of OTC by introducing deletions in the vicinity of the OTC BGC, whole-genome transcriptional analysis showed changes in expression strength of different BGCs located relatively distant from the OTC gene cluster. The most profound increase in expression profile was observed in two BGCs, BGC 21_1 and 22_2, adjacent to each other, a >32-fold increase in average gene transcript abundance. Importantly, BGCs 21_1 and 22_2 were cryptic in the parent ATCC 10970 strain, located 1.6 Mb away from the OTC BGC and, by a single perturbation (the 145 kb deletion), expression was induced to substantial levels. Analyzing the metabolic profile of the engineered strain ATCC Δ145kb, we observed two new

metabolites (longicatenamycin B and longicatenamycin C and encoded by BGC 22_1), which could not be observed by LC/MS analysis of the ATCC 10970 parent strain under identical conditions. Therefore, our results show that a relatively simple and precise deletion of 145 kb in the sub-telomeric region of the *S. rimosus* chromosome can have a profound effect on the expression of diverse BGCs located at different positions on the chromosome, which is in agreement with studies carried by Deng et al. (66).

Our results clearly demonstrate that increase in OTC BGC expression in ATCC Δ145kb strain containing the 145 kb deletion is the reason for massive OTC titer increase. However, an additional copy of the OTC gene cluster introduced in the chromosome "in *trans*" has a profound effect on OTC titer increase of around 30%. As we already demonstrated earlier (29), translocation of the OTC cluster has a positive effect on OTC production due to increased expression of genes presented in OTC BGC, which was translocated (29). As described in our previous observation by Pikl et al. (29), it is not unexpected that the OTC titer increase in the ATCC Δ145 kbΔOTC + ɸC31OTC is boosted by the additional copy of the OTC BGC at this new specific location. Pikl et al. (29) demonstrated that an increase in OTC titer was clearly due to the OTC BGC over-expression phenotype, suggesting that the re-location of the OTC BGC has a substantial effect on the strength of OTC BGC expression. This effect likely contibutes positively to the increased OTC titer in the Δ145kbΔOTC + ɸC31OTC strain.

However, to evaluate the effect of copy number of the OTC BGC on OTC titer, we also introduced the pYAC-ΦC31-Ts-OTC plasmid (29) containing the entire OTC BGC into the ATCC Δ145 kb strain, thus generating a strain containing a second copy of the OTC BGC in the ATCC Δ145 kb + ɸC31OTC strain (see Fig. 10B). Remarkably, the transformant (ATCC Δ145 kb + pYAC-ΦC31-Ts-OTC) had a titer of 5.2 g/L of OTC (Fig. 11), which is almost 10× higher compared to the parent ATCC 10970 strain and almost double the amount (2.8 g/L, Fig. 2) produced by the ATCC Δ145 kb strain. In this experiment, we demonstrated that the addition of a second OTC BGC copy at the ɸC31 *att* site further increased the OTC titer in the ATCC Δ145kb strain. Importantly, the enhanced induction in OTC BGC expression as a result of the introduction of the 145 kb deletion could be enhanced further by the the addition of an exta copy of the OTC BGC.

This result demonstrates that the chromosome location of the BGC plays an important role in antibiotic gene expression. However, we were somewhat surprised that such a significant additional increase of OTC titer of around 30% is observed when the source of carbon is increased only for 15% in the culture of the strain ATCC Δ145 kb + φC31OTC, which is containing the additional copy of the OTC BGC.

Final concentration of the starch in the production medium was increased from 28 to 40 g/L, which is still a very low concentration, compared to the content of the starch used in the industrial production media. Hence, compared to the ATCC strain, over 10-fold increase of the OTC titer was achieved in the medium with moderate increased content of carbon source. Interestingly, the final titer of the engineered strain ATCC Δ145 kb + φC31OTC has almost doubled, compared to the industrial strains M4018 and R6-500. This experiment also suggests, that in the engineered strain ATCC Δ145kb containing 145 kb deletion, the OTC BGC gene expression intensity still did not reach ceiling, and further increase in the gene expression would likely cause significant OTC titer increase.

To conclude, based on our results, larger-scale deletions, amplifications, or more complex chromosomal rearrangements, occurring spontaneously or due to mutagenesis and strain selection, and located at the chromosomal ends where there is a high density of BGCs, can cause drastic changes in metabolic profile. The approach that we applied in the scope of this work demonstrates the potential of introducing targeted large-to-medium-scale deletions as a versatile technology to increase titer of the target secondary metabolite and to activate so-called cryptic gene clusters, which present an enormous source of yet unexplored natural products of medical and industrial value.

## Conclusions

Typically, *Streptomyces* genomes encode 20–60 biosynthetic gene clusters. However, under laboratory conditions, most are either silent or poorly expressed so that their products are only detectable at nanogram quantities, which hampers natural product drug development efforts. Recently, a number of approaches to induce expression of silent BGCs have been applied without significant success. Here, we used comparative genome analysis of industrial *S. rimosus* strains, developed during decades of industrial strain improvement by intensive mutagenesis and selection regimes, to instruct rational strain improvement. We carried out targeted genome deletions in *S. rimosus* ATCC 10970, the progenitor of the industrial strains used in this study. Although large-scale rearrangements have been observed in diverse *Streptomyces* species in the past, most studies related to the regulation of BGCs focussed on regulatory networks. Until recently, actinomycete genome organization, the chromosomal location of BGCs and topology (folding) of the bacterial chromosome were not recognized as an important factor significantly affecting expression of secondary metabolites. We show that a targeted deletion of 145 kb fragment in the vicinity of OTC BGC influences expression of the OTC BGC, as well as others. Genome plasticity of actinomycetes is well known. This phenomenon has important implications in the evolution of actinobacteria in the natural environment. Analogously, decades of industrial strain improvement regimes of commercial *Streptomyces* can also be considered a unique evolutionary process where enhanced antibiotic titer is selected, and our data suggest that large-scale chromosome rearrangements can have an important role in strain improvement. Further studies on genome organization and BGCs expression are needed to better understand the effect of genome topology on gene expression.

## MATERIALS AND METHODS

### Strains and growth conditions

Plasmids were propagated in *Escherichia coli* DH10β, cultured in 2TY medium with apramycin (100 µg/mL) at 28°C. *Escherichia coli* ET12567/pUB307 was used for conjugation and grown in 2TY at 28°C with apramycin (100 µg/mL) and kanamycin (50 µg/mL). All the *Streptomyces rimosus* strains (Table S8) were cultivated on soy-mannitol (MS) agar plates for sporulation and in tryptone soy broth medium (TSB) for genomic DNA isolation and plasmid removal (subcultivations). Throughout all experiments, *S. rimosus* standard growth conditions (SGC) were set at 28°C and 220 rpm, in a rotary shaker. Thiostrepton (30 µg/mL) was used for the selection of *S. rimosus* exconjugants. OTC fermentations were conducted in 50 mL falcon tubes with working volume of 5 mL, shaken at SGC. Seed cultures were prepared by inoculating with an agar plug from a freshly grown culture into 5 mL of GOTC-V medium (29) and incubated for 30 h at SGC. Then, the seed cultures were inoculated into GOTC-P production medium (29) at 10% (vol/vol) and incubated for 5 days in SGC and 60% humidity. When larger volumes of the cultures were needed for transcriptome analysis and preliminary tests, the production phase was moved to 250 mL Erlenmeyer flasks that were filled with 40 mL of GOTC-P medium and inoculated with 4 mL of vegetative stage culture prepared in 50 mL falcon tubes.

### Construction of plasmids

The spacer sequences, plasmids, and primers from this study are listed in Tables S3, S9, and S10, respectively. The primers used for plasmid construction contained 20–36 bp of appropriate overlapping regions to enable different homology-based cloning methods. All spacer sequences were selected with help of the gRNA activity scoring algorithm (67) in Geneious R11.1.5 software (https://www.geneious.com). All homology regions were amplified from genomic DNA isolated from *S. rimosus* ATCC 10970. The Large del_DOWN homology was identical for 145

and 240 kb deletions and was amplified by a set of primers: DOWN_Fw and DOWN_Rw. The upstream homology regions were amplified with corresponding primer pairs: Δ145kb_UP homology: Δ145kb_UP_Fw/Δ145kb_UP_Rw; Δ240kb_UP homology: Δ240kb_UP_Fw/Δ240kb_UP_Rw. gRNA parts were PCR amplified from synthesized fragments (ATG Biosynthetics GmbH; Germany). All synthesized gRNA parts (Δ145kb_gRNA, Δ240kb_gRNA, ΔrimA_gRNA) were composed of two consecutive gRNA cassettes. The single gRNA cassette consisted of synthetic P21 promoter[67] followed by 20 bp spacer sequence and native *S. pyogenes* trans-activating CRISPR RNA and t0 terminator (32).

### pRep_P1_cas9_Δ145kb assembly

The pRep_P1_cas9_tsr plasmid was first linearized with *SpeI* enzyme. The Δ145kb_gRNA, Δ145kb_UP fragments were fused with overlap extension PCR (OE-PCR). The OE-PCR consisted of equimolar amounts of each part, initially run for 10 cycles without primers. After the first step, the external primers (gRNA_FW and Δ145kb_UP_Rw) were added to the reaction and run for another 30 cycles. The Δ145 kb OE-PCR fragment was then joined to the Large del_DOWN PCR fragment and *SpeI*-linearized pREP_P1_cas9_tsr in a homology-based SLiCE reaction (68) to obtain the final plasmid construct pRep_P1_cas9_Δ145kb. pRep_P1_cas9_Δ240kb assembly: first, one of the two *SpeI* restrictions sites on the pRep_P1_cas9_Δ145kb was deleted, to enable *SpeI* digestion without the separation of downstream homology from the plasmid backbone. This was achieved by digesting the pRep_P1_cas9_Δ145kb plasmid with *BstBI* single-cutter, with restriction site adjacent to unwanted *SpeI* site and disruption of the *SpeI* site with NEB builder (New England Biolabs, USA) and primer Stich_delete_SpeI. The reconstituted plasmid was then digested with *SpeI* and *DraI*, resulting in linearized pREP_P1_cas9_ Large del_DOWN plasmid backbone. The gRNA and UP homology parts for 240 kb deletion were fused together with OE_PCR approach and assembled into pRep_P1_cas9_Δ240kb plasmid using SLiCE cloning reaction. pRep_P1_cas9_ΔrimA assembly: PCR amplified ΔrimA_gRNA was fused with ΔrimA_UP and ΔrimA_DOWN parts with OE_PCR approach and assembled into pRep_P1_cas9_ΔrimA plasmid using SLiCE cloning reaction.

## Construction of engineered strains

pRep_P1_cas9_Δ145 kb, pRep_P1_cas9_Δ240 kb, and pRep_P1_cas9_ΔrimA plasmid constructs were first introduced to *E. coli* strain Et-pUB307 (69) that mediated the transfer of the pREP plasmid constructs to *S. rimosus* ATCC 10970 and ATCC 10970 ΔOTC. According to the conjugation procedure, *E. coli* ET-pUB307 carrying pREP-CRISPR plasmids and *S. rimosus* spores were mixed and plated on MS agar plates (40 mL), supplemented with 30 mM MgCl$_2$ and 30 mM CaCl$_2$. After 4–6 h, the plates were overlayed with 1.5 mL aqueous solution of nalidixic acid and thiostrepton, both to final concentration of 30 mg/L. After 6 days, exconjugants were transferred to MS medium containing thiostrepton and nalidixic acid. Viable exconjugants were then inoculated into 5 mL TSB medium and subjected to two rounds of subcultivation (24 h at 28°C and 220 rpm). Subcultivation aimed to remove the replicative pREP-CRISPR-plasmids from the mutant strains to avoid the occurrence of non-specific mutations, caused by prolonged presence of active Cas9-gRNA complex. One hundred microliters of $10^{-6}$ dilution of subcultivated cultures was spread on MS agar plates with and without the selection marker (thiostrepton). Colonies from MS plates where substantial loss of plasmid was observed were further patched onto new replica agar plates. Plasmid loss was evident from inability of colonies to grow in presence of thiostrepton. Verification of deletions was performed as described in Supplementary Methods (Supplementary information 8).

## WGS of ATCC Δ145kb strains

Whole-genome sequencing (WGS) was undertaken with Illumina technology. gDNA was isolated from the two *S. rimosus* ATCC 10970 Δ145kb (a/b) mutants for which the 145 kb deletion had been confirmed by PCR and two ATCC 10970 colonies. Library preparation and DNA sequencing were performed by Macrogen, Inc. (Daejeon, Republic of Korea). The short-read genome-sequencing library was prepared with TruSeq DNA PCR Free kit (Illumina, Inc., San Diego, CA, USA), and sequencing was performed on a Novaseq Illumina platform (Illumina, Inc., San Diego, CA, USA). The short sequence reads obtained from Illumina sequencing were initially subjected to quality control and adapter sequence trimming using BBDuk trimmer (version 38.84) (https://jgi.doe.gov/data-and-tools/bbtools/). The trimmed reads were aligned to the ATCC 10970 reference genome (21). Alignment was performed using the Geneious mapper tool, employing a medium-low sensitivity setting and iterating the process five times to ensure optimal alignment. All these operations were executed within Geneious Prime 2022 1.1 (https://www.geneious.com).

## Culture conditions, sampling, and isolation of total mRNA for RNA seq analysis

For transcriptomic analysis, seed cultures were prepared by inoculating an agar plug from a freshly grown culture of *S. rimosus* strains into 5 mL of GOTC-V medium in a 50 mL falcon tube and incubated for 30 h at 220 rpm, 28°C. For the production stage, 4 mL of seed cultures were inoculated to 40 mL of GOTC-P medium in 250 mL Erlenmeyer flasks and cultivated for 50 h. Cultures were prepared in three replicates. The cells were sampled from culture broth at 24 and 48 h, fixed by adding 5 times the amount of fixative (ethanol: phenol 96:4), and stored at −80°C. Total RNA was isolated with RNeasy Mini Kit (Qiagen), as described in Supplementary Information 6.

## Formatting and analysis of RNA-seq data

Total RNA was isolated and sent to Novogene (China) for Sample QC & Library preparation with 250–300 bp insert strand-specific library with rRNA removal and sequencing with 2 Gb (6.6 M reads) raw data output. After QC checking for GC compliance, RNA-seq library was prepared according to the classic workflow for library construction (Novogene QC Analysis Report). Sequencing was performed on an Illumina NovaSeq (Illumina, USA) in paired-end mode with read length of 150 bp (PE150). Base read quality of obtained sequences was very high, with the lowest Q20 and Q30 sample scores of 97.93% and 94.61%, respectively. Raw paired-end reads obtained from Novogene (China) were trimmed using BBDuk trimmer (version 38.84) to remove adapters and bases not reaching phred score Q15. Trimmed sequences were then simultaneously aligned to reference *S. rimosus* ATCC 10970 chromosome (CP048261) and large linear plasmid (CP048262) accessible at https://www.ncbi.nlm.nih.gov/nuccore/CP048261.1/ using Bowtie2 (70). Expression levels of reference CDS aligned sequences were then determined using embedded Genious Prime quantification tool, where ambiguously mapped reads were counted as partial matches. Differential expression analysis between replicate data of control (ATCC 10970) and mutant strains was performed using DEseq2 (71), which also produced PCA plots (Fig. S8) and volcano plots (Fig. 6; Fig. S4) that were visualized with Genious Prime 2022 1.1 (https://www.geneious.com). Clustering was done with calculated Euclidean distances with average linkage, with the differentially expressed genes (adjusted *P*-value < 0.01; absolute log$^2$FC > 2.5) ordered by maximizing the sum of similarities of adjacent elements. All visualizations and computations related to clustering were carried out in Orange 3.35.0 (72).

## HPLC analysis of OTC production

After fermentation of *S. rimosus* strains in GOTC-P medium (Methods—strains and growth conditions) titers of OTC in fermentation broths were analyzed using HPLC, as previously described by Pikl et al. (29). Briefly, 1 mL of fermentation broth was acidified to pH 1.5–2.0 with 37% HCl and centrifuged at 13,000 rpm for 10 min. The supernatants were filtered with 0.45 µm syringe filters and subjected to HPLC analysis (UltiMate 3000 HPLC system) using a C-18 column (150 × 4.6 mm; 5 µm; 40°C; Macherey-Nagel), with UV absorption detection at 270 nm, and a flow rate of 1 mL/min. The elution solvent A (80% $H_2O$, 20% MeOH, 0.1% formic acid) and solvent B (100% MeOH) were used in 20 min gradient elution as follows: The 0→8 min, 10% B; 8→12 min, 10%→90% B; 12→15 min, 90% B; 15→15.01 min, 90%→10% B; 15.01→20 min, 10% B. The OTC titers were calculated as the mean from at least three parallel samples of each mutant and the parental strains.

## LC-MS and HRMS analysis of the engineered strains

Samples were prepared with five times dilution of water, stirred well, and centrifuged. Acetonitrile extracts were prepared by adding 4 vol. of ACN to culture broths. Supernatants (5 µL) were injected onto a Merck Porospher STAR RP-18 endcapped column (100 × 2.1 mm, 2 µm), with a flow rate of 0.4 mL min$^{-1}$ at 60°C column temperature. A gradient method was performed using mobile phase A: acetonitrile/water/formic acid, 10:990:1 (vol/vol/vol), and mobile phase B: acetonitrile/water/formic acid, 800:200:1 (vol/vol/vol). Starting with 100% mobile phase A (0% mobile phase B), mobile phase B was first brought to 7.6% at 1.2 min, and then to 80% at 9.0 min, before returning back to 0% for re-equilibration of the column for 2 min. MS detection was performed with an LCQ iontrap mass spectrometer (ThermoFisher, USA) equipped with a HESI source. Positive ionization (source voltage 5 kV, heater OFF, capillary temperature 275°C, sheath gas 10 AU, auxiliary gas 5 AU, sweep gas 0 AU) and full-scan monitoring with *m/z* range 150–1,000 allowed the detection of the compounds, primarily as proton [M + H]+ adducts. UV diode array multi-wavelength detection was performed in parallel. Raw data were visualized using QualBrouser, a part of Xcalibur package (ThermoFisher, USA) and MZmine 3.0 (41). HRMS analysis was performed on an UltiMate 3000 UHPLC system (Thermo Scientific, USA) coupled with a triple quadrupole/linear ion trap mass spectrometer (4000 QTRAP LC-MS/MS System; Applied Biosystems/MDS Sciex, Ontario, Canada). Methanol (Chromasolv LC-MS grade, Fluka, Switzerland) and water purified on a Milli-Q system from Millipore (Bedford, MA, USA) were used for the preparation of mobile phases, and formic acid from Fluka was used as modifier. An analytical HPLC column Kinetex XB-C18 100A (2.1 × 100 mm, 2.6 µm particle size, Phenomenex) was used with the flow rate of 0.5 mL min$^{-1}$ (the details on method gradient and Ms experiments are given in supplementary information). A mobile phase consisted of acetonitrile and water, both modified with 0.1% formic acid, was used throughout the work. Injection volume and column temperature were 10 µL and 30°C, respectively. HRMS measurements were performed with a hybrid quadrupole orthogonal acceleration time-of-flight mass spectrometer (QTOF Premier, Waters, Milford, MA, USA).

## Statistical analysis

Statistical analysis of OTC production as quantified by HPLC was performed using GraphPad Prism v9. Comparisons were performed with Dunnett's T3 Test assuming unequal variances ($n = 9$). Corresponding *P* values are reported in figure legends.

### ACKNOWLEDGMENTS

We thank MSc. students Julija Horvat, Tim Godec, and Carmen Díez Acedo for their assistance during the experiments. We also thank Katja Stare and Sara Fišer for their help with RNA isolation.

The study was supported by the Ministry of Higher Education, Science and Technology (Slovenian Research Agency, ARRS, grants No. P4-0116 to H.P., P4-0165 and P1-0034). A.P. is supported by a young researcher grant from the Slovenian Research Agency (No. 53621).

Conceptualization: A.P., L.S., and H.P.; Methodology: A.P., L.S., M.A., M.T., M.Š., P.H., M.S., P.H., M.P., Š.B., D.H., A.S., and P.M.; Investigation: A.P., L.S., M.A., M.T., P.M., M.S., P.H., and H.P.; Resources: H.P., P.H., D.H., A.S.; Data curation: A.P., M.T., M.Š., P.H., M.S., Š.B., and A.S.; Writing—original draft preparation: A.P., M.T., P.H., I.S.H., A.S., and H.P.; Supervision: M.A., P.M., A.S., I.S.H., and H.P.; Project administration: A.P., Š.B., A.S., and H.P.; Funding acquisition: H.P.

All authors read and approved the final manuscript.

## AUTHOR AFFILIATIONS

[1]Chair of Biotechnology, Microbiology and Food Safety, University of Ljubljana Biotechnical Faculty, Ljubljana, Slovenia
[2]National Institute of Chemistry, Ljubljana, Slovenia
[3]Strathclyde Institute of Pharmacy and Biomedical Sciences, University of Strathclyde, Glasgow, United Kingdom
[4]Faculty of Food Technology and Biotechnology, University of Zagreb, Zagreb, Croatia
[5]Educational and Scientific Institute of High Technologies, Taras Shevchenko National University of Kyiv, Kyiv, Ukraine
[6]Department of Biotechnology and Systems Biology, National Institute of Biology, Ljubljana, Slovenia
[7]Antiinfectives, Sandoz, Mengeš, Slovenia

## PRESENT ADDRESS

Miha Tome, National Institute of Biology, Ljubljana, Slovenia

## AUTHOR ORCIDs

Alen Pšeničnik ⓘ http://orcid.org/0000-0001-5375-2220
Miha Tome ⓘ http://orcid.org/0000-0001-7729-7381
Paul Herron ⓘ http://orcid.org/0000-0003-3431-1803
Marko Petek ⓘ http://orcid.org/0000-0003-3644-7827
Špela Baebler ⓘ http://orcid.org/0000-0003-4776-7164
Hrvoje Petković ⓘ http://orcid.org/0000-0003-1377-9845

## FUNDING

| Funder | Grant(s) | Author(s) |
| --- | --- | --- |
| Javna Agencija za Raziskovalno Dejavnost RS (ARRS) | P4-0116 | Hrvoje Petković |
| Javna Agencija za Raziskovalno Dejavnost RS (ARRS) | P4-0165 | Špela Baebler |
| Javna Agencija za Raziskovalno Dejavnost RS (ARRS) | P1-0034 | Martin Šala |
| Javna Agencija za Raziskovalno Dejavnost RS (ARRS) | 53621 | Alen Pšeničnik |

## AUTHOR CONTRIBUTIONS

Alen Pšeničnik, Conceptualization, Formal analysis, Investigation, Methodology, Software, Visualization, Writing – original draft, Writing – review and editing | Lucija Slemc, Investigation, Methodology | Martina Avbelj, Investigation, Methodology | Miha Tome, Conceptualization, Data curation, Formal analysis, Methodology, Software, Visualization, Writing – review and editing | Martin Šala, Investigation, Methodology | Paul Herron, Conceptualization, Data curation, Methodology, Software, Writing – original draft, Writing – review and editing | Maksym Shmatkov, Data curation, Formal analysis,

Software | Marko Petek, Writing – review and editing | Špela Baebler, Data curation, Methodology, Writing – review and editing, Investigation | Peter Mrak, Formal analysis, Investigation, Methodology, Writing – review and editing | Daslav Hranueli, Conceptualization | Antonio Starčević, Software, Writing – review and editing | Iain S. Hunter, Project administration, Software, Writing – review and editing | Hrvoje Petković, Conceptualization, Funding acquisition, Project administration, Resources, Supervision, Writing – original draft, Writing – review and editing

## DATA AVAILABILITY

All relevant data generated or analyzed during this study are included in this published article and its supplementary information files. The datasets generated and/or analyzed during the current study are available from the corresponding author on reasonable request. Complete RNA-seq data and genome sequencing reads of R6-500, M4018, and mutant strains are available at NCBI BioProject PRJNA1031826. Materials will be made available for non-profit research under a material-transfer agreement upon reasonable request to the corresponding author.

## ADDITIONAL FILES

The following material is available online.

### Supplemental Material

**Data S1 (mSystems00250-24-s0001.xlsx).** GO enrichment analysis of RNA-seq data.
**Data S2 (mSystems00250-24-s0002.pdf).** Results of RNA-seq clustering analyis: heatmap (Fig. 5) with marked gene locus tags.
**Additional experimental details (mSystems00250-24-s0003.docx).** Supplemental information, figures, and tables.

### Open Peer Review

**PEER REVIEW HISTORY (review-history.pdf).** An accounting of the reviewer comments and feedback.

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
