## [Reviewer comments · mSystems]

Oxytetracycline hyper-production through targeted genome reduction of *Streptomyces rimosus*

Alen Pšeničnik, Lucija Slemc, Martina Avbelj, Miha Tome, Martin Šala, Paul Herron, Maksym Shmatkov, Marko Petek, Špela Baebler, Peter Mrak, Daslav Hranueli, Antonio Starčević, Iain Hunter, and Hrvoje Petković

Corresponding Author(s): Hrvoje Petković, Univerza v Ljubljani Biotehniška fakulteta

Review Timeline:

Submission Date:

February 20, 2024

Accepted:

March 7, 2024

Editor: Zackery Bulman

Reviewer(s): The reviewers have opted to remain anonymous.

Transaction Report:

DOI: <https://doi.org/10.1128/msystems.00250-24>

Re: mSystems00250-24 (Oxytetracycline hyper-production through targeted genome reduction of *Streptomyces rimosus*)

Dear Prof. Hrvoje Petković:

Thank you for the opportunity to review your manuscript. Your manuscript has been accepted, and I am forwarding it to the ASM production staff for publication. Your paper will first be checked to make sure all elements meet the technical requirements. ASM staff will contact you if anything needs to be revised before copyediting and production can begin. Otherwise, you will be notified when your proofs are ready to be viewed.

Cover Image Submissions: If you would like to submit a potential Cover Image, please email a file and a short legend to msystems@asmusa.org. Please note that we can only consider images that (i) the authors created or own and (ii) have not been previously published. By submitting, you agree that the image can be used under the same terms as the published article. Image File requirements: TIF/EPS, 7.5 inches wide by 8.25 inches tall (at least 2,250 pixels wide by 2,475 pixels tall), minimum 300 dpi resolution (600 dpi preferred), RGB, and no figure elements, e.g., arrows or panel labels. The legend should be a short description of the image, 1-2 sentences recommended.

Sincerely,
Zackery Bulman
Editor

mSystems

Reviewer #2 (Comments for the Author):

The authors have addressed my previous concerns. I have no further comments.